# SEE IT TO PLACE IT: EVOLVING MACRO PLACEMENTS WITH VISION LANGUAGE MODELS

## ABSTRACT

We propose using frontier Vision-Language Models (VLMs) for macro placement in chip floorplanning, a complex optimization task that has recently shown promising advancements through machine learning methods. For human designers, macro placement is an inherently visual process that relies on spatial reasoning to arrange components on the chip canvas. Because VLMs exhibit strong reasoning capabilities over visual inputs, we hypothesize that these models can effectively complement existing learning-based approaches. We introduce VeoPlace (Visual Evolutionary Optimization Placement), a novel framework that uses a VLM to guide the actions of a base policy by constraining them to subregions of the chip canvas. The VLM proposals are iteratively optimized through an evolutionary search strategy with respect to resulting placement quality. On open-source benchmarks, VeoPlace establishes a new state-of-the-art for learning-based methods, outperforming the strongest prior approach across all evaluated circuits by reducing wirelength by an average of 10.9% with peak improvements of over 20%. Our approach opens new possibilities for electronic design automation tools that leverage foundation models to solve complex physical design problems.

## 1 INTRODUCTION

Computer chip floorplanning is a critical step in the integrated circuit design process, involving the strategic arrangement of macros on the chip canvas. Determining the optimal placement is a complex multi-objective problem, in which performance, power, and area (PPA) must be optimized while respecting constraints such as routing congestion. The vast combinatorial design space makes manual chip floorplanning a time-consuming and expertise-driven task. For example, placing $M$ macros on an $N \times N$ grid creates a design space of size $O(N^{2M})$, rendering exhaustive search infeasible.

A variety of approaches have been proposed for automated chip floorplanning, including black-box optimization (Shi et al., 2023), analytical methods (Lin et al., 2019; Cheng et al., 2018; Lu et al., 2015), and learning-based methods (Mirhoseini et al., 2021; Lai et al., 2022; 2023; Lee et al., 2024). Among these, learning-based approaches have achieved state-of-the-art performance, but have a severe limitation: policies trained from scratch struggle to generalize to *unseen chips* without additional interaction, an issue exacerbated by the limited training data available in chip design. In contrast, human designers leverage high-level prior knowledge and spatial reasoning to efficiently tackle new design spaces. Our work aims to bridge this gap by harnessing frontier Vision-Language Models (VLMs) to provide human-like spatial reasoning and guide the exploration of existing learned models.

Existing learning-based approaches first pre-train models on a set of training chips, then fine-tune on sampled placements from unseen chips (Mirhoseini et al., 2021; Lai et al., 2022; 2023). We consider a general formulation of this setup: for an unseen block and a fixed budget of $B$ placement attempts, what is the best possible placement that can be generated? We posit that efficiently using this budget of online attempts requires spatial reasoning as well as learning from prior attempts - capabilities that have been both exhibited by modern VLMs.

Our proposed method, VeoPlace (Visual Evolutionary Optimization Placement), is structured hierarchically, and uses a high-level VLM planner to guide a low-level placement policy by constraining it to promising regions. Crucially, these VLM proposals are iteratively refined through an evolutionary process, as visualized in Figure 1. VeoPlace requires no fine-tuning of the VLM (we use

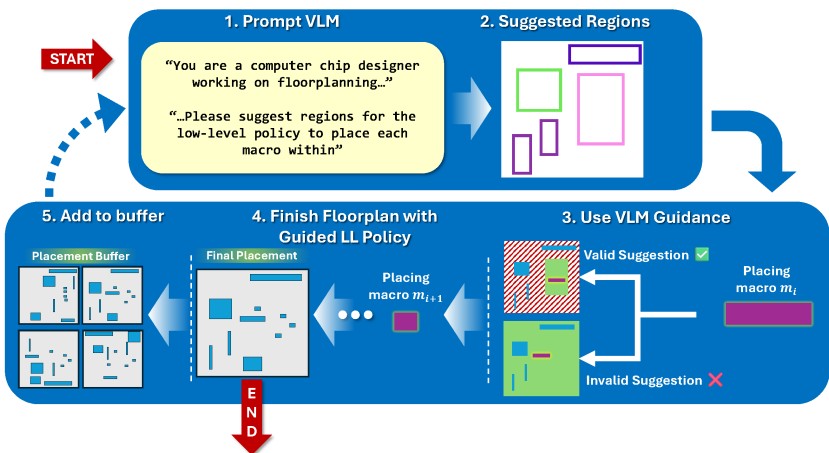

Figure 1: VeoPlace framework overview. The VLM suggests placement regions (1-2) to constrain a low-level policy (3) for macro placement (4). A history buffer that stores the existing population of placements (5) facilitates evolutionary in-context improvement, creating a feedback loop to improve placement quality.

the public Gemini models (Team et al., 2023)) and uses an independently trained low-level policy (ChiPFormer (Lai et al., 2023)). On open-source benchmarks, VeoPlace establishes a new state-of-the-art for learning-based methods by reducing wirelength by an average of **10.9%** against the state-of-the-art prior work, with peak improvements exceeding 20%. Our main contributions are:

- **Novel VLM-guided placement framework**: We introduce VeoPlace, the first framework to show that foundation models can guide specialized placement algorithms via spatial reasoning, without requiring fine-tuning.

- **Evolutionary context selection strategy**: A novel context selection mechanism that enables VLMs to iteratively improve placement quality by focusing on geometrically similar high-performing solutions, outperforming diverse and random selection strategies.

- **Inverse scaling in context length**: We identify a surprising inverse scaling behavior of VeoPlace performance with respect to context length, and provide key evidence that even state-of-the-art VLMs under-explore when provided with more examples in-context.

## 2 RELATED WORK

**Automated Chip Floorplanning.** Automating computer chip floorplanning has been studied through various approaches (Table 1), including analytical methods (Lin et al., 2019), black-box optimization techniques such as simulated annealing (Wong & Liu, 1986) and genetic algorithms (Singha et al., 2012), more recent guided black-box methods (Shi et al., 2023), and various learning-based methods (Mirhoseini et al., 2021; Lai et al., 2023; Geng et al., 2024b). Within the learning-based category, a prominent line of work formulates chip floorplanning as a reinforcement learning (RL) problem where macros are sequentially placed onto a chip canvas (Mirhoseini et al., 2021; Lai et al., 2022; 2023). Alternatively, recent works have proposed learning to refine existing chip floorplans, employing techniques such as diffusion models (Lee et al., 2024) or RL algorithms (Xue et al., 2024) for post-processing. Our approach can be viewed as a generalization of learning-based approaches (and explicitly leverages them in the inner loop) by using a high-level VLM to guide them at test-time.

**Vision-Language Models for Decision-Making.** Vision-Language Models (VLMs) are trained on vast datasets of text and images, and therefore contain rich priors valuable for tasks requiring both vision and language (Driess et al., 2023). The use of VLMs to perform decision-making has been explored in several fields, including robotics, where VLMs interpret natural language commands within a visual scene to guide robot actions or planning (Ahn et al., 2022; Jiang et al., 2022; Shridhar et al., 2022; Huang et al., 2022; Brohan et al., 2023; Kim et al., 2024; Team et al., 2025; Liang et al., 2024). Systems such as SayCan (Ahn et al., 2022) and RT-2 (Brohan et al., 2023) demonstrate

how VLMs can translate high-level instructions into actionable plans that low-level controllers can execute.

Table 1: Comparison of existing macro placement approaches. Our VLM+IL approach represents a novel direction in the field. RL: Reinforcement Learning, MCTS: Monte Carlo Tree Search, IL: Imitation Learning, BBO: Blackbox Optimization, VLM: Vision-Language Model, LLM: Large Language Model.

| Method | Category |
|---|---|
| SP-SA (Murata et al., 2002) | Packing |
| NTUPlace3 (Chen et al., 2008) | Analytical |
| RePlace (Cheng et al., 2018) | Analytical |
| DREAMPlace (Lin et al., 2019) | Analytical |
| GraphPlace (Mirhoseini et al., 2021) | RL |
| DeepPR (Cheng & Yan, 2021) | RL |
| MaskPlace (Lai et al., 2022) | RL |
| EfficientPlace (Geng et al., 2024a) | RL + MCTS |
| ChiPFormer (Lai et al., 2023) | IL |
| WireMaskBBO (Shi et al., 2023) | BBO |
| EvoPlace (Yao et al., 2025) | LLM + Analytical |
| VeoPlace (Ours) | VLM + IL |

Our work leverages these VLM capabilities in a similar hierarchical approach. The VLM perceives chip placement images along with their performance metrics, analyzes spatial arrangements, and provides high-level guidance to a low-level policy in the form of suggested bounding regions for each macro. These bounding regions constrain the action space for the low-level policy, effectively creating a division of labor where the VLM handles high-level spatial reasoning while a specialized policy executes precise placement decisions within these constraints.

**Pairing LLMs with Evolution.** Pairing LLMs with evolutionary search has achieved success in fields such as program generation (Romera-Paredes et al., 2024; Hemberg et al., 2024; Liventsev et al., 2023), planning and reasoning (Lee et al., 2025), scientific discovery (Yamada et al., 2025; Gottweis et al., 2025), robotics (Nasiriany et al., 2024), and chip design (Novikov et al., 2025; Yao et al., 2025; Xue et al., 2024; Shi et al., 2023). VeoPlace adopts an evolutionary framework where the VLM acts as a generator of new solutions (region proposals). These solutions are evaluated, and high-performing ones are selected to inform the VLM's next generation of proposals, given feedback from an objective function. Our selection strategy is to focus on a population of high-performing, geometrically similar placements. That is, VeoPlace explicitly focuses on evolution in a local region, a principle that has been shown to be effective in sparse Gaussian processes (Wei et al., 2024) and island models in genetic algorithms (Romera-Paredes et al., 2024; Lee et al., 2025; Tanese, 1989; Cantú-Paz et al., 1998).

## 3 PRELIMINARIES

**Macro Placement.** We consider macro placement in chip floorplanning, where a set of *macros* $M = \{m_1, \ldots, m_N\}$, defined by their dimensions and connectivity, are placed on a 2D chip *canvas*. Connectivity is given by a *netlist* $G = (M, E)$, a hypergraph where each hyperedge (*net*) connects a subset of macros. The objective is to find a placement $P = \{p_1, \ldots, p_N\}$, where $p_i$ is the bottom-left corner of macro $m_i$, that minimizes estimated wirelength. This is the total length of wiring needed to connect the components (macros and standard cells) within each net, and is a crucial metric for a chip's performance, power, and area (PPA) (Lin et al., 2019; Mirhoseini et al., 2021).

Macro placement is formulated as a sequential decision-making problem, a Markov Decision Process (MDP) (Mirhoseini et al., 2021; Lai et al., 2023; 2022). In this setup, macros are sequentially placed onto the canvas, typically following a predefined order such as descending macro area. The state $s_t$ encompasses information about the current partial placement (locations of macros $m_1, \ldots, m_{t-1}$), features of the current macro $m_t$, and potentially structural information derived from the netlist $G$.

To manage the continuous placement space, the canvas is discretized into a grid of cells, where an action $a_t$ selects a specific grid cell for the reference point (e.g., the bottom-left corner) of the current macro, $m_t$. After all $N$ macros are placed, a terminal reward $R$ is computed based on the final Half-Perimeter Wirelength (HPWL). The total HPWL is the sum of the half-perimeters of the smallest axis-aligned bounding box for each net in the netlist $G$. The agent's goal is to learn a policy $\pi(a_t|s_t)$ that maximizes the expected terminal reward $\mathbb{E}[R]$ (or, equivalently, minimizes HPWL).

**Reward.**   Our agent is responsible for placing macros, but the final design quality is heavily influenced by the vast number of small standard cells. To guide the agent, we use a reward composed of two parts: (1) a dense, per-step reward during the episode based on the incremental wirelength estimation from MaskPlace (Lai et al., 2022) and (2) a terminal reward calculated based on the placement of the remaining standard cells. This terminal reward is the `grouped-HPWL` computed by using the DREAMPlace analytical placer (Lin et al., 2019) to position the standard cells around the agent's fixed macro placements. Our objective differs from that of (Lai et al., 2023) in the inclusion of this terminal reward term, which significantly improves correlation with the true objective. Following (Mirhoseini et al., 2021), we cluster the standard cells using hMETIS in order to speed up standard cell optimization. As we demonstrate in Section C.2, the grouped-HPWL strongly correlates with the true global HPWL on most benchmarks, validating its use as a reliable reward component (DREAMPlace hyperparameters are detailed in Section D.3).

**Inference-time Optimization.**   Online RL requires many environment interactions and model updates to produce optimal placements for new netlists. Recent work suggests that offline RL pre-training provides strong zero-shot performance but benefits from fine-tuning on a small amount of online interaction (Lai et al., 2023). We consider an inference-time optimization setting that uses a **hierarchical approach**: we are allowed a fixed budget of placement evaluations, but do not fine-tune either the VLM (our high-level strategic guide) or the low-level policy. As Section 5 shows, VeoPlace can achieve results superior to fine-tuning, suggesting greater efficiency on new tasks.

**ChiPFormer.**   We adopt ChiPFormer (Lai et al., 2023) as our low-level policy—responsible for the task of selecting a specific grid cell for each macro—due to its state-of-the-art performance and multi-task generality. It is an autoregressive Transformer model trained with an offline Decision Transformer objective (Chen et al., 2021; Lee et al., 2022). Crucially for our framework, ChiPFormer outputs a probability distribution over grid cells for each macro. While ChiPFormer can operate standalone to provide a competitive baseline, it lacks the high-level visual and spatial reasoning capabilities like human designers. Our VLM-based guidance addresses this limitation by directly constraining ChiPFormer's action distribution, steering the policy toward better design choices and improving placement quality without requiring any fine-tuning of the VLM or ChiPFormer.

## 4   VEOPLACE

In this section, we describe VeoPlace, our novel evolutionary framework that harnesses the spatial reasoning of VLMs for chip floorplanning. The framework iteratively evolves a population of placements, using a VLM as a variation operator. The VLM generates region proposals based on prior attempts, which in turn constrain the rollouts of a low-level placement policy. As illustrated in Figure 1, the VLM suggests promising regions (steps 1–2) that serve as spatial constraints for the low-level policy as it generates a complete floorplan (steps 3–4). High-performing placements are stored in a history buffer and fed back to the VLM (step 5), creating an evolutionary feedback loop that continuously improves placement quality.

### 4.1   VLM AND LOW-LEVEL POLICY INTERFACE

VeoPlace is an inference-time evolutionary search strategy that orchestrates the interaction between a VLM and a stochastic low-level placement policy $\pi$ that parameterizes a probability distribution over locations to place the next macro. As shown in Algorithm 1, VeoPlace iteratively evolves a population $H$ of placements (i.e., episodes) by rolling out the low-level policy with VLM proposals and generating new VLM proposals based on prior attempts and the corresponding placement outcomes. In each episode (Algorithm 1, line 3), a complete placement $P_e$ is generated by sequentially placing macros $m_t$ from the netlist $G$. Following ChiPFormer, we roll out the low-level policy to generate candidate placements, but additionally leverage a VLM to provide high-level guidance to this placement process. As outlined in Algorithm 1 (lines 5-7), VeoPlace mixes low-level-only rollouts with VLM-guided rollouts (every $K$ episodes).

When querying the VLM, we provide a `context` derived from the existing population in the history buffer $H$ of previously generated floorplans (see Section 4.3 for details) and the current netlist $G$.

Based on this input context, the VLM suggests bounding box regions $\{s_1, ..., s_n\}$ on the chip canvas for the respective macros (visualized in Figure 4).

---

**Algorithm 1** VeoPlace

---

**Require:** V: Vision-language model
 $\pi$: Stochastic ChiPFormer policy
 G: Netlist with macros $\{m_1, ..., m_n\}$
 $C$: Context length
 $E$: Total number of episodes
 $K$: Interval to query the VLM
1: Initialize placement population $H \leftarrow \emptyset$
2: **for** episode $e = 1$ to $E$ **do**
3:     Initialize placement $P_e \leftarrow \emptyset$
4:     **if** $e \bmod K = 0$ **then**
5:         context $\leftarrow$ BUILD_CONTEXT$(H, C)$
6:         $\{s_1, s_2, ..., s_n\} \leftarrow V(\text{context}, G)$ {Sample suggestions for all macros}
7:     **end if**
8:     **for** macro $m_t \in \{m_1, m_2, ..., m_n\}$ **do**
9:         **if** suggestion $s_t$ for $m_t$ is valid **then**
10:             $p_t \sim \pi(\cdot|m_t, P_e, s_t)$ {Use suggested region}
11:         **else**
12:             $p_t \sim \pi(\cdot|m_t, P_e)$ {Use original policy}
13:         **end if**
14:         $P_e \leftarrow P_e \cup \{(m_t, p_t)\}$ {Update placement}
15:     **end for**
16:     Calculate $HPWL_e$ for placement $P_e$
17:     $H \leftarrow H \cup \{(P_e, HPWL_e)\}$ {Update population}
18: **end for**
19: **return** $H$

---

VeoPlace then rolls out the low-level policy, but constrains its actions at each timestep $t$ to the suggested region $s_t$, denoted by $\pi(\cdot|m_t, P_e, s_t)$ (Algorithm 1, line 10). This is practically achieved by masking the policy's output logits outside of $s_t$ before sampling. This constrains $\pi$ to place $m_t$ within the area identified as promising by the VLM, while still retaining control over the exact placement coordinates within the region.

Because the low-level policy autoregressively places macros, a VLM suggestion $s_t$ may be invalid for any $t > 1$ due to already-placed macros overlapping the region. In this case, the macro $m_t$ is placed by sampling from the original, unconstrained policy distribution, $p_t \sim \pi(\cdot|m_t, P_e)$ (Algorithm 1, line 12). Finally, after each placement $P_e$ is completed, its quality (e.g., $HPWL_e$) is calculated and the pair $(P_e, HPWL_e)$ is added to the population $H$ (Algorithm 1, lines 18-19).

## 4.2 STRUCTURED PROMPT

VeoPlace prompts a VLM to generate bounding box suggestions for each macro $\{s_1, ..., s_n\}$ conditioned on previous placements and their evaluations $\{P_i\}$. We generate suggestions for all macros simultaneously to reduce VLM inference. The prompt's key characteristics are (1) its structure, which elicits spatial reasoning, and (2) the selection of in-context examples (detailed in Section 4.3, with a full example in Appendix F.1).

We find that VLMs struggle with macro placement due to information overload and a lack of domain-specific knowledge, often producing inconsistent or imprecise spatial suggestions without proper guidance (see Appendix F.3). Our structured prompt guides the VLM with clear objectives, constraints, and a standardized format (Table 2), transforming its general visual reasoning into useful spatial guidance. To ensure generalizability and avoid overfitting, we developed our prompt exclusively on the adaptec1 benchmark, applying the final version to all other benchmarks without modification.

Table 2: Components of the structured prompt for VeoPlace.

| Component | Description |
|---|---|
| **Grid Representation** | • Discrete 84x84 grid canvas, matching ChiPFormer (Lai et al., 2023) resolution. |
| | • Macros positioned by their bottom-left corner coordinates. |
| | • Coordinate system origin (0,0) at the bottom-left. |
| **Visual Representation** | • Image of the canvas showing all currently placed macros and their colors. |
| | • Provides visual context for spatial relationships, available space, and patterns (see Fig. 4). |
| **Context Elements** | • Grid specifications and current macro properties (dimensions, color). |
| | • A history of prior placement attempts with their performance metrics. |
| | • The current state (locations) of all previously placed macros. |

## 4.3 CONTEXT SELECTION STRATEGIES FOR EVOLUTION

The core component of our evolutionary algorithm is prompting the VLM to generate a superior placement suggestion given a set of prior placements and their evaluations. Because each placement

uses hundreds of tokens, only a small number can be provided to the model while maintaining reasonable inference cost. Given this limited budget, the examples should be (1) high-quality, so the model improves upon good placements, and (2) informative enough for the model to effectively deduce better placements via reasoning. We consider the following candidate strategies for selecting a fixed context of $C$ examples:

**Most Recent (FIFO)**: Select the $C$ most recently generated placements from the population. This strategy implements pure evolutionary search, where the VLM observes a temporally ordered sequence of recent attempts.

**Random**: Randomly sample $C$ placements from the population buffer. This provides a baseline that makes no assumptions about which examples are most valuable for evolutionary search.

**Best Performing**: Select the $C$ placements with the lowest grouped-HPWL from the population to encourage the VLM to replicate successful patterns.

**Diverse**: Represent each placement as a vector in $\mathbb{R}^{2T}$ by concatenating the $(x_i, y_i)$ coordinates of its $T$ macros. We then perform K-means clustering with $C$ clusters on the population and select the placement with the best grouped-HPWL from each cluster. This promotes geometric diversity across the selected examples while still favoring high-quality designs.

**Top Stratified**: Represent placements as coordinate vectors and cluster them by geometric similarity, as in the diverse strategy. We then focus on a single promising cluster by ranking all clusters by their best grouped-HPWL and sampling one using a softmax distribution over these ranks (with probability proportional to $e^{-i/\tau}$ for rank $i$). From the selected cluster, we choose the top $C$ performing layouts, supplementing from nearby clusters if needed. This strategy provides a set of geometrically similar examples that represent variations of a particular design pattern.

These context-building strategies represent competing hypotheses for optimizing the evolutionary search. Our findings in Section 5 reveal that minimizing geometric diversity among examples (**Top Stratified**) yields the best performance. This suggests that consistency in exemplars helps the VLM identify and apply relevant placement patterns, effectively allocating the limited context budget to the most promising search region (Wei et al., 2024).

# 5 EXPERIMENTS

We design experiments to address two key questions: **(Q1)** Can VLM guidance improve the placement quality of a learning-based policy using only inference-time computation? **(Q2)** What are the key underlying factors of the inference-time strategy (such as prompt, context size, and context selection) that influence the final performance of VeoPlace?

**Setup.** Our evaluation setup follows Lai et al. (2023), using open-source chip benchmarks from the ISPD 2005 (Nam et al., 2005) and ICCAD 2004 (Adya & Markov, 2002; Adya et al., 2004) challenges. These benchmarks vary in complexity, with hundreds to thousands of macros and up to hundreds of thousands of standard cells, providing a broad test across different scales and designs. We use a pre-trained ChiPFormer model as the low-level policy in all experiments that was trained using the public repository [1]. Further details are provided in Appendix D.2.2.

Our setup differs from that of Lai et al. (2023) in two main aspects. (1) We train the base ChiPFormer policy with an additional sparse reward from the DREAMPlace analytical placer to better align the baseline with final, mixed-size placement quality. We report only these mixed-size placement metrics (as opposed to also macro-only metrics) because these more accurately determine the final quality of the placement. To enable fast mixed-size placement rewards at inference-time, we cluster the standard cells following Mirhoseini et al. (2021). We validate this proxy metric `grouped-HPWL` in Section C.2, confirming its strong correlation with global HPWL and its reliability as an optimization target. (2) To match prior work such as Mirhoseini et al. (2021), we fix the locations of all macros placed by ChiPFormer during DREAMPlace. Empirically, we find that allowing movable macros results in significant changes to their final placements, which confounds the actual efficacy of the base policy (see Figure 5).

---

[1] https://github.com/laiyao1/chipformer

Table 3: Comparison of VeoPlace (VP) and ChiPFormer (CF). We compare two VP variants, which use a context length of one ($C = 1$) and do not fine-tune the low-level policy, against CF with and without fine-tuning (FT). We report mean grouped Half-Perimeter Wirelength (`grouped-HPWL`, $\times 10^7$, lower is better) with standard deviation across three random seeds, reflecting the best performance achieved by each method over 4,000 rollouts. The best result for each benchmark is bolded.

| | VP 2.0 Flash ($C = 1$) | VP 2.5 Flash ($C = 1$) | ChiPFormer | |
| --- | --- | --- | --- | --- |
| **Benchmark** | | | No-FT | FT |
| adaptec1 | 7.16±0.17 | **7.11±0.14** | 8.40±0.12 | 8.39±0.19 |
| adaptec2 | 10.76±0.27 | **10.20±0.29** | 13.96±0.18 | 13.07±0.00 |
| adaptec3 | **18.58±0.44** | 18.82±0.20 | 20.11±0.26 | 19.93±0.38 |
| adaptec4 | 17.17±0.06 | **16.76±0.18** | 18.76±0.22 | 18.65±0.09 |
| bigblue1 | 8.76±0.02 | **8.68±0.07** | 9.33±0.05 | 9.21±0.03 |
| bigblue2 | 16.30±0.03 | **16.29±0.04** | 16.65±0.07 | 16.74±0.04 |
| bigblue3 | 43.29±0.44 | **42.38±3.85** | 45.06±0.52 | 47.10±0.58 |
| bigblue4 | **80.07±1.11** | 81.17±2.31 | 93.29±0.23 | 95.91±0.79 |
| ibm01 | 0.31±0.00 | **0.27±0.01** | 0.34±0.00 | 0.34±0.00 |
| ibm02 | **0.58±0.00** | 0.60±0.01 | 0.68±0.00 | 0.65±0.00 |
| ibm03 | 0.66±0.00 | **0.64±0.00** | 0.72±0.01 | 0.71±0.01 |
| ibm04 | **0.76±0.02** | 0.80±0.01 | 0.83±0.01 | 0.82±0.01 |

**VeoPlace Rollout Procedure**   Our method generates an equal number of guided and unguided rollouts in each batch. Specifically, each batch consists of 8 episodes from directly sampling the base policy, and 8 episodes guided by VLM suggestions. We note that Gemini batch inference incurred a median inference latency of 40 seconds for Gemini 2.0 Flash and 200 seconds for Gemini 2.5 Flash (see Appendix Table 4 for Gemini parameters).

## 5.1   Q1: Does VLM Guidance Improve Placement Quality?

Our evaluation against ChiPFormer baselines demonstrates that VeoPlace consistently improves placement quality. We compare against two strong baselines under the same 4,000-rollout budget: a version without fine-tuning (No-FT) that repeatedly samples the fixed ChiPFormer policy, and a version with ChiPFormer fine-tuning (FT) that adapts the policy via online decision transformer. As detailed in Table 3, VeoPlace outperforms both, reducing `grouped-HPWL` by an average of **10.9%** over the best ChiPFormer baseline, with pronounced gains on benchmarks like adaptec2 (22.0%) and ibm01 (20.6%). This performance gap is likely because the FT strategy greedily refines the best placements seen so far, whereas VeoPlace uses the VLM's reasoning for better exploration of the design space.

## 5.2   Q2: What are the key aspects of VLM guidance?

Beyond the VLM itself and its sampling hyperparameters, VeoPlace is also parameterized by the prompt to the VLM. We study three key aspects of this prompt: 1) the base instruction, 2) the number of in-context examples, and 3) the manner in which the in-context examples are selected.

### 5.2.1   Base Instruction

We show how the prompt's base instruction impacts VLM performance by testing two variants in an ablation study: a **greedy** prompt that instructs the VLM to make only minor modifications to existing placements, and an **exploration-focused** prompt that encourages it to generate novel placements.

A clear trend emerges (depicted in Table 10): the exploration-focused prompt outperforms the greedy prompt, confirming that merely exploiting known solutions is insufficient for achieving state-of-the-art results.

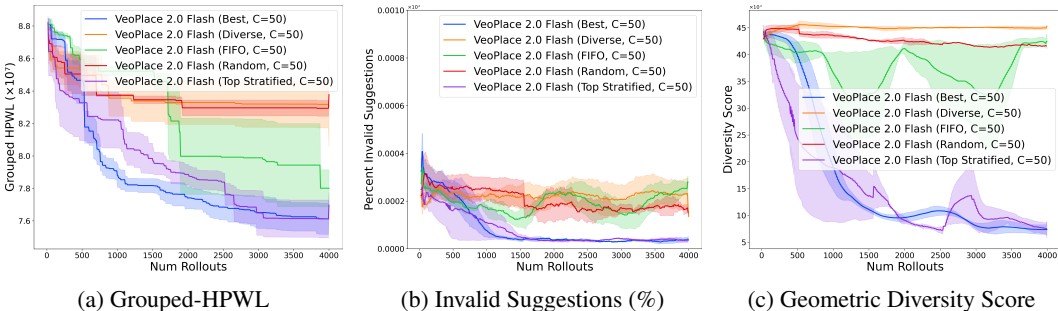

|(a) Grouped-HPWL|(b) Invalid Suggestions (%)|(c) Geometric Diversity Score|

Figure 2: Comparison of context selection strategies on the `adaptec1` benchmark, showing (a) final placement quality (`grouped-HPWL`), (b) VLM valid suggestion rate, and (c) geometric diversity of the context.

### 5.2.2 CONTEXT EXAMPLE SELECTION

The strategy for selecting $C$ in-context examples is a critical component of our evolutionary framework. Because the VLM's context window is limited, the choice of which prior placements to show the model directly impacts its ability to generate high-quality suggestions. This presents competing hypotheses: should the context maximize diversity to encourage broad exploration, or should it minimize diversity to provide a consistent, focused signal for refinement? To answer this question, we compare five strategies: Most-Recent (FIFO), Random, Best-Performing, Diverse (maximizing geometric distance via clustering), and our proposed Top Stratified approach (minimizing geometric distance within a high-performing cluster).

Our results, presented in Figure 2 for the `adaptec1` benchmark, show a clear trend. Figure 2a shows that the strategies which explicitly minimize geometric diversity (Top Stratified and Best) achieve the lowest final grouped-HPWL. Conversely, strategies that are either diverse by design or randomly sampled perform significantly worse.

The reason for this performance difference is twofold. (1) Figure 2c confirms that the geometric diversity scores of the in-context examples for the Top Stratified and Best strategies are indeed the lowest. (2) This reduced diversity directly improves the VLM's reliability. As shown in Figure 2b, the less geometrically diverse strategies result in fewer invalid suggestions from the VLM. This indicates that the VLM can effectively learn and generalize from a consistent set of high-quality, geometrically similar examples rather than from a diverse set of examples. Based on these findings, we conclude that our **Top Stratified** strategy, which focuses on a high-performing, geometrically similar cluster of examples, is the most effective. We therefore retain this strategy for our main experiments but use a context size of $C = 1$, which provides the VLM with a single example from a stochastically chosen high-quality cluster, a decision we validate further in the following section.

### 5.2.3 REDUCING IN-CONTEXT EXAMPLES IMPROVES PERFORMANCE

We find that for our task, a larger context negatively impacts the VLM by causing two issues: (1) reduced exploration, where the VLM explores less of the placement design space, and (2) impaired context referencing, where the VLM cannot correctly reference specific placements in the context. This results in a surprising inverse scaling behavior where smaller context lengths consistently perform better than larger ones, as shown in Section E.3. For instance, on the `adaptec1` and `bigblue1` benchmarks, the VLM performs significantly better with small contexts (e.g., $C = 1$ or $C = 10$) than with large contexts ($C = 100$ or $C = 150$). Hence, although prior work (Agarwal et al., 2024) has observed positive scaling in the in-context supervised learning setting, we demonstrate an inverse scaling behavior in the in-context reinforcement learning setting.

To investigate this reduced exploration, we visualize the design space explored by the VLM. We represent each placement as a high-dimensional vector of its macro coordinates and use Principal Component Analysis (PCA) to project these vectors into a two-dimensional space. This allows for a qualitative assessment where convex hulls outline the boundaries of the explored regions. As shown in Figure 3b, the area of these convex hulls shrinks as context length increases, confirming that a

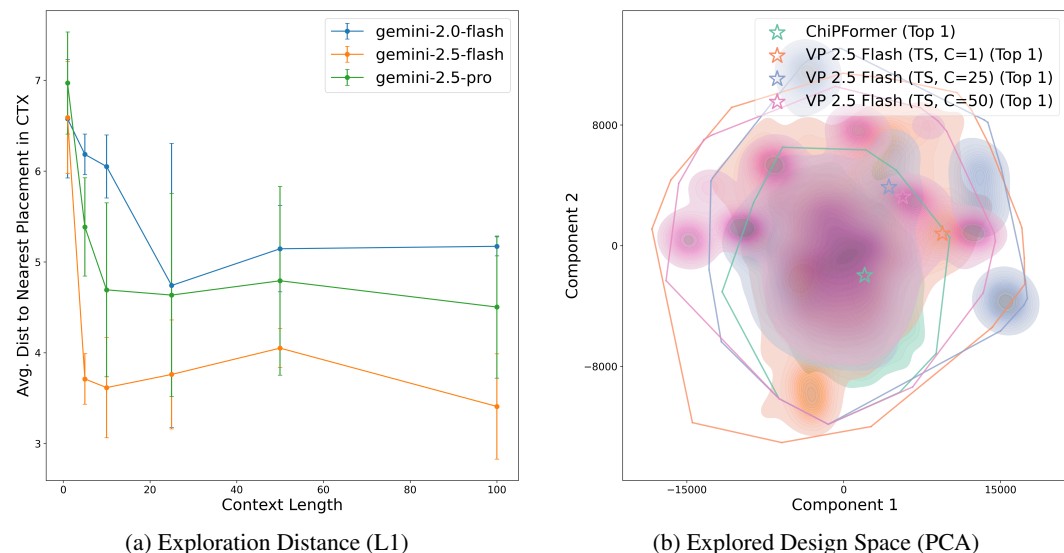

(a) Exploration Distance (L1)    (b) Explored Design Space (PCA)

Figure 3: **Smaller context length ($C$) unlocks broader exploration.** (a) The L1 distance from the suggested placement to the nearest in-context example decreases as context length grows, indicating less exploration. (b) A PCA visualization of the design space shows qualitatively how a smaller context leads to a wider search.

longer context restricts exploration. To quantify this effect, we calculated the average L1 distance between the VLM's suggested macro placements and the geometrically closest placement within its context (Figure 3a). The results show that as context length increases, this average distance decreases for all model variants, confirming that larger contexts lead to less exploration.

We also investigated if impaired context referencing was a primary cause of this performance degradation. Our analysis in Figure 9c shows that while Gemini 2.0 Flash struggles with placement recall as the context length increases, more advanced models like Gemini 2.5 Flash and Pro maintain high accuracy. Since all models exhibit performance degradation with longer contexts regardless of their recall ability, we conclude that reduced exploration is the dominant factor driving the inverse scaling behavior.

## 6 CONCLUSION

We proposed VeoPlace, an evolutionary framework that leverages vision-language models to enhance chip floorplanning. Through a structured prompting approach that requires no domain-specific training, VeoPlace exceeds the performance of state-of-the-art learning-based methods. More importantly, our work provides a blueprint for integrating VLMs into complex engineering workflows as high-level strategic guides. Our findings provide strong evidence that their value lies in reasoning over small, curated sets of high-quality examples, rather than in processing extensive historical data.

Building on this foundation, future work can enhance VeoPlace by incorporating explicit netlist properties, improving computational efficiency, and moving beyond off-the-shelf models to fine-tuned VLMs or language-conditioned policies. Ultimately, this synthesis of general-purpose AI with specialized tools paves the way for more accessible and powerful computer-aided design, with applications extending well beyond chip floorplanning.

ETHICS STATEMENT

The primary application of this research is in electronic design automation (EDA), a field aimed at creating more efficient and powerful integrated circuits. The potential societal benefits include advancements in computing technology and reduced energy consumption from better-designed chips. We acknowledge the computational cost and associated energy consumption required to utilize large-scale Vision-Language Models (VLMs) like Gemini. Our work aims to make this process more efficient by using VLMs as high-level guides in an inference-time optimization loop, rather than requiring costly fine-tuning for each new design. The datasets used are established, open-source academic benchmarks from the chip design community, and our work does not involve human subjects, personal data, or raise fairness and bias concerns. We have adhered to the ICLR Code of Ethics throughout this research.

REPRODUCIBILITY STATEMENT

To ensure the reproducibility of our work, we commit to releasing our source code for the VeoPlace framework upon publication. Our implementation is built upon publicly available models and benchmarks. The low-level policy is a pre-trained ChiPFormer model, trained using the public repository cited in the paper. The high-level VLM planner utilizes the public Google Gemini API. The chip floorplanning benchmarks (ISPD 2005 and ICCAD 2004) are standard and publicly accessible. A comprehensive breakdown of all hyperparameters, including those for the Gemini API, standard cell grouping, and the DREAMPlace analytical placer, is provided in Section D. The core algorithm is detailed in Section 4 (with pseudocode in Algorithm 1), and the structured VLM prompt, a key component of our method, is fully detailed with examples in Section F.1.

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

# Appendix

## Table of Contents

## A LLM USAGE

We used large language models (LLMs) as writing assistants during the preparation of this manuscript. Their role was confined to improving the clarity, grammar, and overall readability of the text. All core research contributions, including the ideation of the method, experimental design, implementation, and analysis of results, were conceived and executed entirely by the human authors, who take full responsibility for all content in this paper.

## B ADDITIONAL DETAILS

### B.1 VISUAL EXAMPLE

To provide a qualitative illustration of our method, Figure 4 visualizes a complete VLM-guided rollout on the `adaptec4` benchmark. This example highlights the hierarchical division of labor central to VeoPlace: the VLM provides a high-level spatial strategy, and the low-level policy executes the precise placement decisions within that strategy. The figure shows the initial VLM proposals ($t = 0$), the state mid-placement ($t = T/2$), and the final floorplan ($t = T$) that results from this guided process.

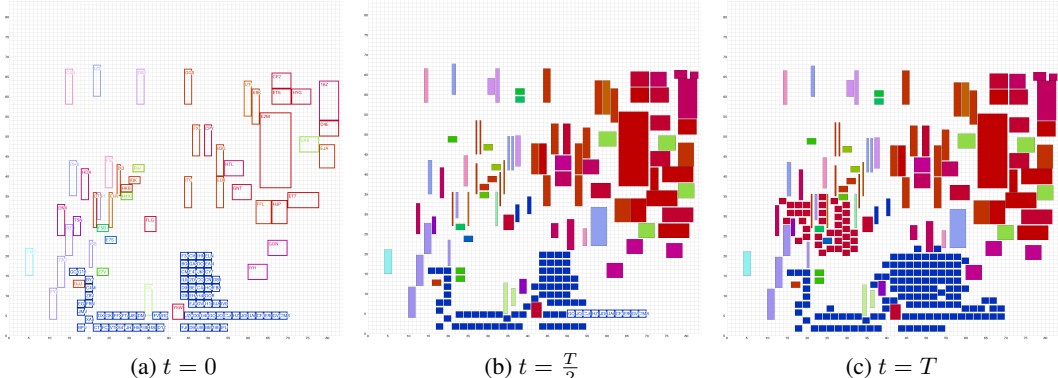

(a) $t = 0$       (b) $t = \frac{T}{2}$       (c) $t = T$

Figure 4: VeoPlace's VLM-guided placement on adaptec4. (a) VLM proposes initial regions ($t = 0$); policy is unconstrained for macros without valid suggestions. (b) Mid-placement ($t = T/2$). (c) Final placement ($t = T$), with the policy operating within VLM constraints.

## B.2 MACRO COLORING

We employ a color-coding strategy to implicitly convey functional relationships between macros to the VLM. This process involves several steps:

First, we construct a **macro-connectivity graph** from the original netlist $G$. In this graph, nodes (excluding standard cells) represent the macros to be placed. An undirected, weighted edge is created between any two macros if they share one or more nets in $G$. The weight of such an edge is proportional to the number of nets these two macros commonly share. This construction effectively flattens the hypergraph structure of the netlist into a standard graph, where indirect connections through nets are represented as weighted between macros.

This macro-connectivity graph is embedded into a low-dimensional space (specifically, an 8-dimensional space in our implementation for k-means) using a spring-based graph layout algorithm. Such algorithms, like the one implemented in the NetworkX Hagberg et al. (2008) Python library, position macros in the embedding space such that those with stronger connections in the graph are located closer to one another in space.

With macros represented as points in this embedding space, we apply k-means clustering to group them. To determine a suitable number of clusters, $k$, we iterate through a predefined range of potential $k$ values (e.g., from 2 to 30). For each $k$, we perform k-means clustering and evaluate the resulting cluster separation using the Silhouette score Rousseeuw (1987). The value of $k$ (and its corresponding clustering) that yields the highest Silhouette score is selected as optimal.

Finally, macros are assigned colors based on their cluster membership: all macros within the same cluster receive the same unique color. Any macros that are not part of the main connectivity graph (e.g., isolated macros not sharing nets with other considered macros, if any) are assigned a default gray color. While specific netlist connectivity details are not directly fed to the VLM, this color-coding, derived from the underlying circuit structure, provides a strong visual heuristic for potential functional groupings and spatial affinities.

## C EXPERIMENTAL SETUP

### C.1 IMPACT OF MOVABLE MACROS IN ANALYTICAL PLACEMENT

In our main evaluation, we report results with fixed macro placements, where the analytical placer only positions the standard cells around the macros (a two-stage flow). This differs from some prior work that allows the analytical placer to move all cells, including macros (a three-stage flow). We find that allowing macros to be movable during the analytical placement stage results in substantial displacements from their original VeoPlace-generated locations. This confounds the evaluation, as the final placement is no longer representative of VeoPlace's learned spatial reasoning but rather the output of the analytical placer.

Figure 5 provides a clear visual demonstration of this effect on the `adaptec1` benchmark. The significant displacement of macros is evident, justifying our decision to use a fixed-macro evaluation to ensure we are measuring the direct efficacy of VeoPlace.

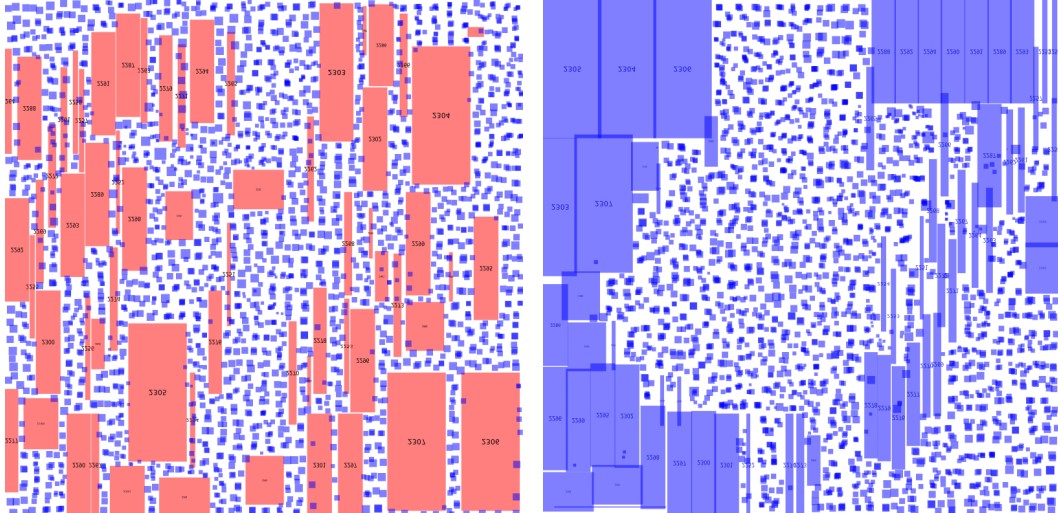

(a) Our two-stage flow (VeoPlace with fixed macros).     (b) Alternative three-stage flow (movable macros).

Figure 5: Visual comparison demonstrating why a fixed-macro flow is essential for a fair evaluation of VeoPlace. (a) The final layout from our two-stage flow, where macro placements generated by VeoPlace are **fixed**. (b) The result of an alternative three-stage flow, which takes the **exact same initial placement** from (a) as input but allows DREAMPlace to move all cells. The analytical placer's drastic rearrangement of macros in (b) shows that it effectively ignores the initial solution, making its final metrics an invalid measure of VeoPlace's contribution. Our fixed-macro approach ensures a direct and unconfounded evaluation.

## C.2    Wirelength Correlation Study

To validate our use of grouped-HPWL as a proxy reward, we measured its correlation with the true global HPWL across all benchmarks. For designs with a smaller number of standard cells (e.g., `adaptec1` with 211k standard cells), the correlation is very strong ($R > 0.9$), confirming that grouped-HPWL is an excellent surrogate for the wirelength objective. For larger benchmarks such as `bigblue3` (1.1M standard cells) and `bigblue4` (2.2M standard cells), the correlation weakens or breaks down. This primarily reflects the fixed DREAMPlace hyperparameters that we used for consistency across experiments (e.g., number of bins, iterations). In practice, these parameters could be tuned for each circuit to sharpen the reward signal, but we did not pursue this here since our focus is to demonstrate the proof-of-concept value of VLM-guided placement rather than fully optimize the surrogate metric. Overall, the strong correlation on the majority of benchmarks supports our use of the grouped proxy as the reward for both the VLM and RL policy.

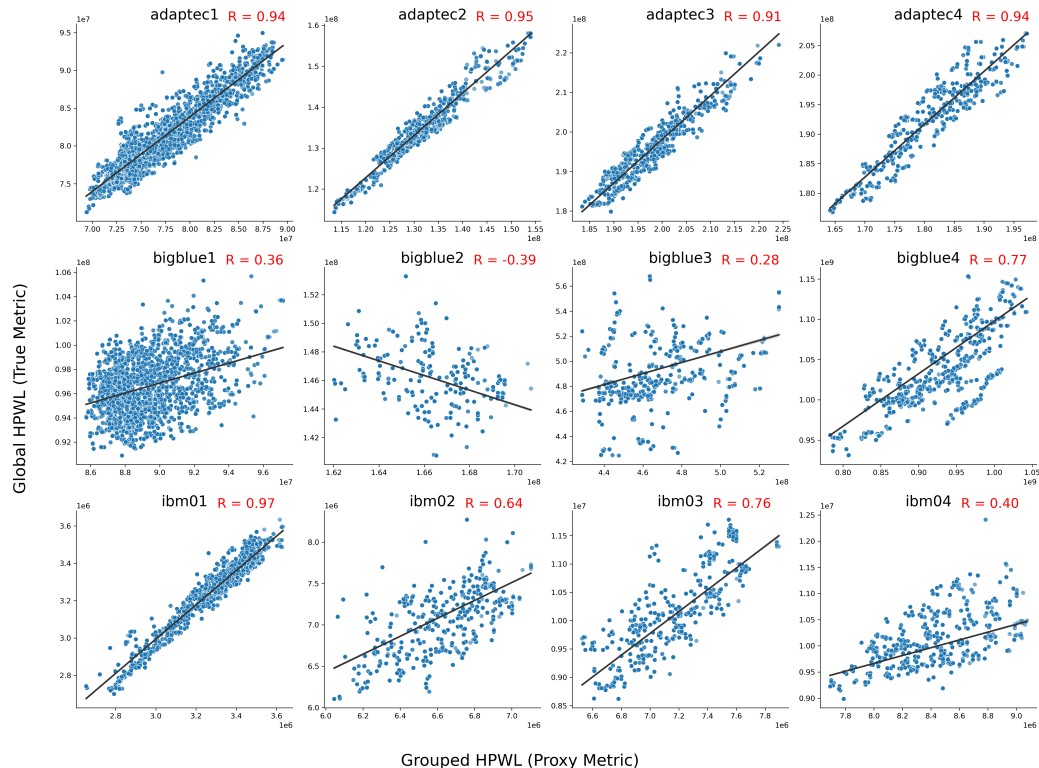

Figure 6: Correlation between grouped-HPWL and global HPWL across benchmarks. Each subplot pools all random seeds for a single netlist, with red text indicating the Pearson coefficient $R$.

## C.3 MODEL SIZE

The ChiPFormer decision transformer model contains approximately 3 million trainable parameters.

## C.4 PRE-TRAINING

The pre-training phase was conducted on servers equipped with 4× NVIDIA A100 40GB GPUs.

## C.5 ROLLOUT AND INFERENCE

For generating rollouts, the computational requirements were significantly lower as these involved only forward passes through the trained 3M parameter model. These were conducted on a single A100 GPU or equivalent, with each benchmark circuit rollout typically completing within seconds.

## C.6 VLM INTEGRATION

For VLM integration, we used Google's Gemini API with the Gemini 2.0 Flash, Gemini 2.5 Flash, and Gemini 2.5 Pro models. Our experiments are organized into iterations, where each iteration consists of 8 rollouts. These iterations alternate: one iteration is generated using only the low-level policy, and the next is generated with VLM guidance.

A VLM query is performed once at the beginning of each guided iteration. Over a 4000-rollout experiment, this results in 500 total iterations (250 unguided and 250 guided), leading to exactly 250 calls to the VLM. Each call to Gemini returns 8 candidate generations, with each generation being a complete set of suggested regions for all macros in the netlist.

## D    HYPERPARAMETERS

### D.1    GEMINI

We use the public Gemini API endpoint for our experiments. Table 4 shows these hyperparameters.

Table 4: Gemini API hyperparameters.

| Parameter | Value |
|-----------|-------|
| Temperature | 0.7 |
| Top-$k$ | 64 |
| Top-$p$ | 0.95 |
| Candidates | 8 |

### D.2    STANDARD CELL GROUPING PARAMETERS

As discussed in Section 5, our proxy metric for placement quality, grouped-HPWL, relies on grouping the hundreds of thousands of standard cells into a smaller set of clusters. This is a crucial preprocessing step that makes the wirelength estimation computationally tractable during the evolutionary search.

We use the open-source codebase from Google's Circuit Training project for this task, which implements the grouping methodology first introduced by Mirhoseini et al. (2021). To ensure a consistent basis for comparison, we applied the same set of grouping hyperparameters across all benchmarks used in our experiments. The specific values for these parameters are detailed in Table 5.

Table 5: Hyperparameters for the Standard Cell Grouping Algorithm.

| Parameter | Value | Description |
|-----------|-------|-------------|
| Number of Groups | 2000 | The fixed number of clusters to group all standard cells into. |
| Cell Area Utilization | 1.25 | A target for the density of cells within a cluster. |
| Enable Group Breakup | True | A boolean flag that allows the algorithm to split larger groups. |

#### D.2.1    PRETRAINING

**Circuit Tokens**    For pretraining the circuit token representation component using the Variational Graph Auto-Encoder (VGAE), we used the following hyperparameters:

- Hidden layer dimensions: [32, 32]
- Learning rate: 0.01
- Training epochs: 800

**Transformer**    Following ChiPFormer (Lai et al., 2023), we use a reward-conditioned transformer with the following hyperparameters:

- Number of transformer layers: 6
- Number of attention heads: 8
- Embedding dimension: 128

#### D.2.2    ROLLOUT SETTINGS

**Returns-to-go**    We configured specific target returns-to-go for each benchmark netlist to guide the generated placements. Since our objective is to minimize wirelength, we define the reward as its negative value. Following the methodology of Decision Transformers, we set these values to ambitious targets, encouraging the model to generate high-quality placements with very low

wirelengths. Table 6 shows the target returns-to-go values used for each benchmark circuit in our experiments.

Table 6: Target Returns-to-Go for Different Benchmark Netlists

| Netlist | Return-to-go |
|---------|--------------|
| adaptec1 | -9.20E+07 |
| adaptec2 | -1.15E+08 |
| adaptec3 | -2.09E+08 |
| adaptec4 | -1.82E+08 |
| bigblue1 | -9.35E+07 |
| bigblue2 | -1.64E+08 |
| bigblue3 | -5.14E+08 |
| bigblue4 | -8.35E+08 |
| ibm01 | -6.59E+06 |
| ibm02 | -9.39E+06 |
| ibm03 | -1.08E+07 |
| ibm04 | -1.16E+07 |

## D.3 DREAMPLACE

### D.3.1 GROUPED STANDARD CELLS

The hyperparameters used for DREAMPlace when placing the grouped standard cells are detailed in Table 7. All experiments utilize DREAMPlace version `4.2.0`.

Table 7: DREAMPlace Hyperparameters for Grouped Standard Cells

| Netlist | Target Density | Stop Overflow | Density Weight | Num Bins (X) | Num Bins (Y) | Iterations |
|---------|---------------|---------------|----------------|--------------|--------------|------------|
| adaptec1 | 1.00 | 0.07 | $8 \times 10^{-5}$ | 64 | 64 | 500 |
| adaptec2 | 0.75 | 0.07 | $8 \times 10^{-5}$ | 64 | 64 | 500 |
| adaptec3 | 0.75 | 0.07 | $8 \times 10^{-5}$ | 64 | 64 | 500 |
| adaptec4 | 1.00 | 0.07 | $8 \times 10^{-5}$ | 64 | 64 | 500 |
| bigblue1 | 1.25 | 0.07 | $8 \times 10^{-5}$ | 64 | 64 | 500 |
| bigblue2 | 1.25 | 0.07 | $8 \times 10^{-5}$ | 64 | 64 | 500 |
| bigblue3 | 1.25 | 0.07 | $8 \times 10^{-5}$ | 64 | 64 | 500 |
| bigblue4 | 0.50 | 0.07 | $8 \times 10^{-5}$ | 64 | 64 | 500 |
| ibm01 | 0.75 | 0.07 | $8 \times 10^{-5}$ | 64 | 64 | 500 |
| ibm02 | 0.50 | 0.07 | $8 \times 10^{-5}$ | 64 | 64 | 500 |
| ibm03 | 0.50 | 0.07 | $8 \times 10^{-5}$ | 64 | 64 | 500 |
| ibm04 | 0.50 | 0.07 | $8 \times 10^{-5}$ | 64 | 64 | 500 |

# E  ADDITIONAL EXPERIMENTS

## E.1  HPWL COMPARISON

This section presents the experimental results specific to each of the twelve netlists used in our evaluation. The figures below illustrate the performance characteristics of our algorithm across the different circuit designs in terms of grouped Half-Perimeter Wirelength (`grouped-HPWL`), measured in units of $10^7$. Lower HPWL values indicate better placement quality with reduced interconnection length, demonstrating the effectiveness of our placement strategy across varying netlist complexities and structures.

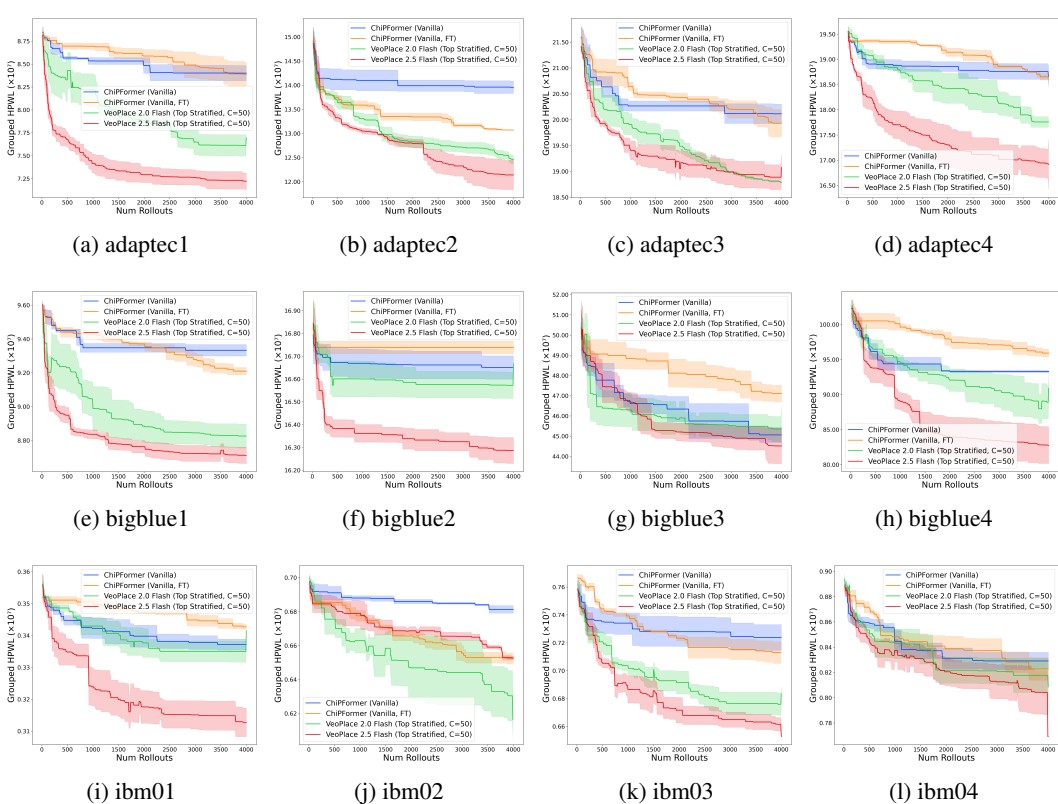

Figure 7: Grouped Half-Perimeter Wirelength (HPWL $\times 10^7$) results for individual netlists used in our evaluation. Lower values indicate better placement quality with reduced interconnection length.

## E.2  DESIGN SPACE EXPLORATION

To better understand the benefits of VLM guidance, we visualize the design space explored by VeoPlace and contrast it with the space explored by ChiPFormer alone. We represent each placement as a high-dimensional vector of its macro coordinates and use Principal Component Analysis (PCA) to project these vectors into a two-dimensional space. This allows for a qualitative assessment of the search patterns. To directly link exploration with placement quality, we pair these visualizations with box plots of the final Grouped-HPWL metric.

Figure 8 presents this combined analysis across the `bigblue1`, `bigblue2`, and `bigblue3` benchmarks. The left column displays the PCA projections, where convex hulls outline the boundaries of the regions explored by ChiPFormer, VeoPlace 2.0 Flash, and VeoPlace 2.5 Flash (both using Top Stratified with $C = 50$). The right column shows the corresponding Grouped-HPWL distributions for all placements generated by each method. This side-by-side comparison visually demonstrates the relationship between a method's search breadth and its resulting performance.

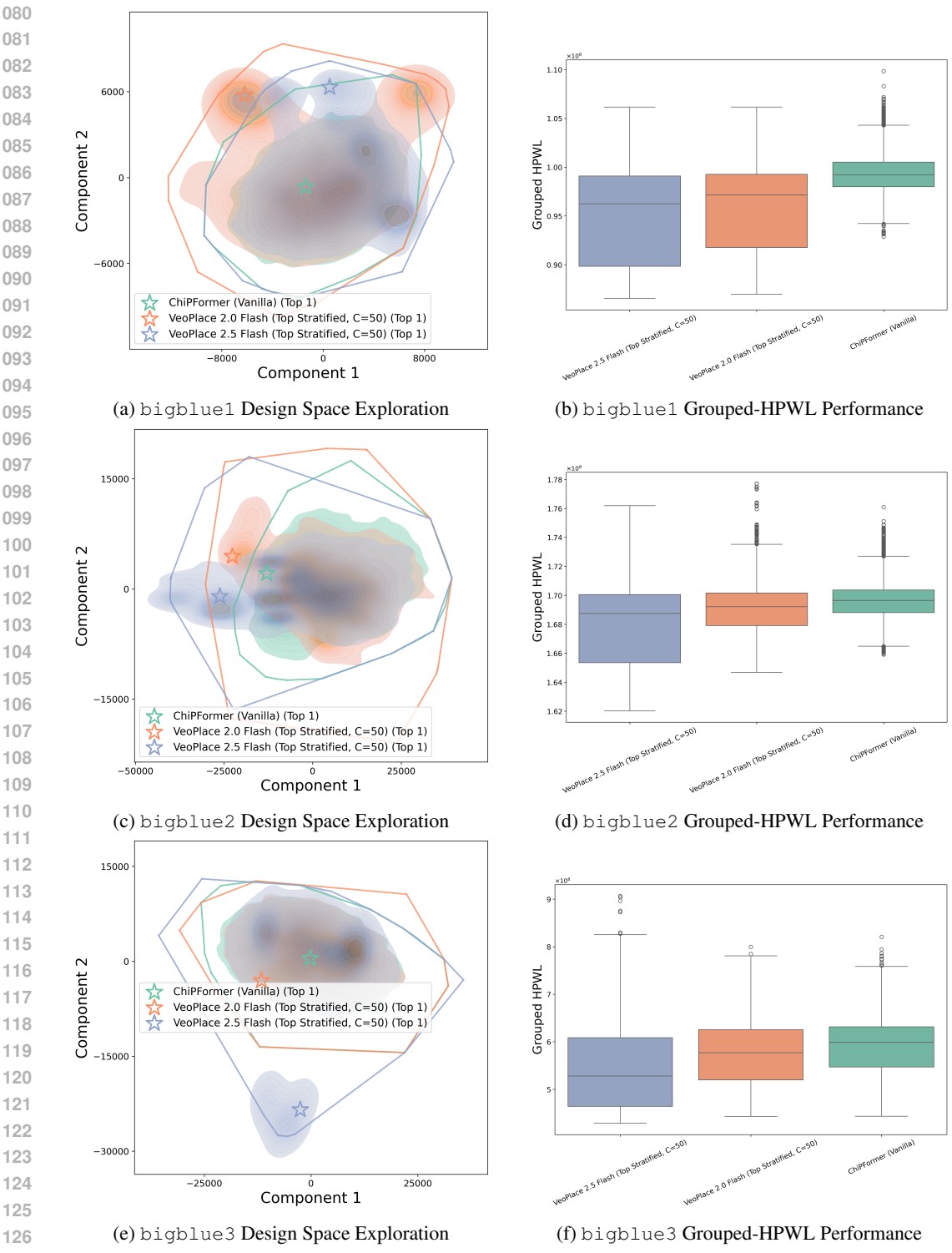

(a) `bigblue1` Design Space Exploration

(b) `bigblue1` Grouped-HPWL Performance

(c) `bigblue2` Design Space Exploration

(d) `bigblue2` Grouped-HPWL Performance

(e) `bigblue3` Design Space Exploration

(f) `bigblue3` Grouped-HPWL Performance

Figure 8: Comparison of design space exploration and placement performance. **Left Column:** PCA visualization of explored design spaces for ChiPFormer (Vanilla), VeoPlace 2.0 Flash, and VeoPlace 2.5 Flash (both Top Stratified, C=50). Convex hulls show exploration boundaries, and stars mark the best placement. **Right Column:** Corresponding boxplots of Grouped-HPWL. The figure illustrates that the broader exploration of VeoPlace variants consistently leads to better placement quality (lower Grouped-HPWL).

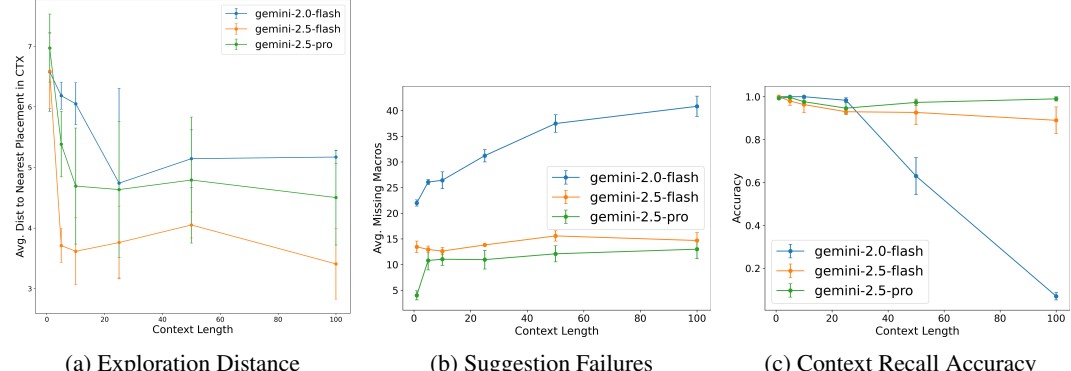

(a) Exploration Distance     (b) Suggestion Failures     (c) Context Recall Accuracy

Figure 9: **Analysis of VLM behavior with varying context length ($C$) on the `adaptec1` benchmark.** Increasing context length leads to reduced exploration (a). It also impairs context referencing, evidenced by an increase in suggestion failures (b) and a sharp decline in recall accuracy (c), highlighting the negative impact of information overload.

### E.3 CONTEXT LENGTH ABLATION

### E.4 PROMPT ABLATION

To isolate the impact of the prompt's high-level strategic guidance, we conducted an ablation study to test the sensitivity of the VLM's performance to its core instructions. A key question is whether the VLM performs best when asked to explore novel design configurations or to greedily refine known, high-quality solutions.

To investigate this, we created two strategic variants of our main prompt: a **Greedy** prompt that explicitly instructs the VLM to make only minor modifications to the best-performing examples provided in-context, and an **Exploratory** prompt that encourages the VLM to disregard prior examples and generate creative, novel placements. Figure 10 shows the key textual differences between these three prompt strategies.

The results of this ablation, presented in Table 10 for three representative benchmarks, demonstrate that the prompt's strategic intent has a significant impact on performance. The **Exploratory** prompt consistently discovers the best layouts for the more complex `bigblue1` benchmark and for the Gemini 2.5 Pro model, highlighting the value of using the VLM to drive design space exploration. While the more balanced **Default** prompt also performs exceptionally well, the **Conservative** prompt is clearly suboptimal, confirming that merely exploiting known solutions is insufficient for achieving state-of-the-art results. This reinforces our conclusion that the VLM's most effective role in the VeoPlace framework is that of a high-level strategic guide tasked with discovering superior geometric arrangements.

Figure 10: Visual comparison of the core instructional text for the three prompt strategies tested in our ablation study. The key differences in strategic guidance are highlighted in bold. The **Greedy** prompt encourages refinement of known good solutions, the **Default** prompt focuses on general optimization, and the **Exploratory** prompt explicitly asks for novel and diverse configurations. This qualitative difference in instruction directly leads to the quantitative performance differences shown in Table 10.

| Greedy Prompt | Default Prompt | Exploratory Prompt |
|---|---|---|
| You are guiding a low-level placement policy for computer chip floorplanning. Your goal is to **refine existing high-quality placements**. The previous examples provided are high-quality solutions. Your task is to suggest regions that are **VERY SIMILAR** to these successful examples. **Do not deviate significantly**. The goal is to **exploit the known good solutions, not explore new ones**. | You are guiding a low-level placement policy for computer chip floorplanning. Your primary goal is to create the **most optimal chip floorplan possible** that minimizes wirelength. Your task is to suggest rectangular regions for placing macros on the chip canvas... Your suggestions should be **highly precise and optimal**. | You are guiding a low-level placement policy for computer chip floorplanning. Your primary goal is to **EXPLORE NOVEL design configurations** to discover new, optimal floorplans that minimize wirelength. The previous examples provided are for context only... **Do not be constrained by them**. Your task is to suggest **diverse and creative** rectangular regions... **Prioritize novelty and exploration** to find potentially superior placements. |

Table 10: Ablation study on the VLM prompt's strategic guidance, conducted on three representative benchmarks with a context size of C=50. We compare two prompt variants: a **Greedy** prompt that instructs the VLM to make conservative refinements, and an **Exploratory** prompt that encourages novel placements. The results consistently show that the exploratory prompt yields the best wirelength (HPWL, $\times 10^7$, lower is better), demonstrating the importance of prompting for design space exploration. The best result for each model and benchmark is **bolded**.

| Benchmark | Method | Greedy Prompt | Exploratory Prompt |
|---|---|---|---|
| adaptec1 | VP 2.0 Flash | $7.56 \pm 0.28$ | $\mathbf{7.50 \pm 0.08}$ |
| | VP 2.5 Flash | $7.77 \pm 0.14$ | $\mathbf{7.31 \pm 0.10}$ |
| | VP 2.5 Pro | $8.09 \pm 0.15$ | $\mathbf{7.19 \pm 0.07}$ |
| | ChiPFormer (Baseline) | $8.40 \pm 0.12$ | |
| bigblue1 | VP 2.0 Flash | $7.97 \pm 0.12$ | $\mathbf{7.92 \pm 0.25}$ |
| | VP 2.5 Flash | $8.12 \pm 0.05$ | $\mathbf{7.82 \pm 0.04}$ |
| | VP 2.5 Pro | $8.36 \pm 0.21$ | $\mathbf{8.18 \pm 0.06}$ |
| | ChiPFormer (Baseline) | $9.33 \pm 0.05$ | |
| ibm01 | VP 2.0 Flash | $0.32 \pm 0.01$ | $\mathbf{0.32 \pm 0.00}$ |
| | VP 2.5 Flash | $0.33 \pm 0.01$ | $\mathbf{0.31 \pm 0.01}$ |
| | VP 2.5 Pro | $0.33 \pm 0.00$ | $\mathbf{0.31 \pm 0.01}$ |
| | ChiPFormer (Baseline) | $0.34 \pm 0.00$ | |

# F PROMPT DETAILS

## F.1 EXAMPLE PROMPT

---

**Prompt example: Default**

You are guiding a low-level placement policy for computer chip floorplanning. Your primary goal is to create the most optimal chip placement possible that minimizes wirelength and cost. Your task is to suggest rectangular regions for placing macros on the chip canvas, which has been divided into a grid. The low-level policy will choose the exact placement location within your suggested regions. Your suggestions should be highly precise and optimal. If there is a macro in the netlist that you are not providing a suggestion for, the low-level policy will place that macro by itself.

The macros are grouped by colors based on their connectivity in the netlist graph, where macros with higher interconnectivity (more pin connections between them) are assigned similar colors. Your goal is to provide optimal region suggestions that will result in the best possible chip placement with minimal wirelength and cost.

This is a global optimization task where you need to consider:

- The impact of your suggested regions on macros that will be placed in the future
- The overall arrangement of the selected macros that minimizes wirelength and cost

**MACRO NAMES AND PROPERTIES FOR THIS NETLIST:**

| Macro | Color | WxH |
|---|---|---|
| FD4 | #9b69e6 | 2 x 18 |
| CXC | #8f45da | 11 x 24 |
| HKU | #8f45da | 11 x 24 |
| FZ6 | #8f45da | 11 x 24 |
| CWI | #8f45da | 11 x 24 |
| EIO | #8f45da | 6 x 24 |
| JXA | #8f45da | 5 x 18 |
| V8F | #8f45da | 5 x 18 |
| G1F | #8f45da | 5 x 18 |
| IJS | #8f45da | 5 x 18 |
| JPT | #8f45da | 5 x 18 |
| DU2 | #8f45da | 5 x 18 |
| J6X | #8f45da | 5 x 18 |
| HJ5 | #8f45da | 5 x 18 |
| 0IL | #ef90df | 5 x 18 |
| FIF | #ef90df | 5 x 18 |
| E6W | #ef90df | 5 x 18 |
| ELG | #ef90df | 5 x 18 |
| HDJ | #a0ef90 | 5 x 18 |
| DSU | #9b69e6 | 5 x 18 |
| G25 | #a0ef90 | 5 x 18 |
| IOQ | #9b69e6 | 5 x 18 |
| KV6 | #efef90 | 5 x 15 |
| IYX | #8f45da | 9 x 7 |
| IIC | #8f45da | 9 x 7 |
| F87 | #8f45da | 7 x 9 |
| GVY | #8f45da | 7 x 9 |
| ISA | #8f45da | 7 x 9 |
| GJ6 | #8f45da | 7 x 9 |
| FIY | #a0ef90 | 3 x 19 |
| PEJ | #9b69e6 | 3 x 19 |

| Macro | Color | WxH |
|---|---|---|
| JQ5 | #8f45da | 3 x 18 |
| EE4 | #8f45da | 3 x 18 |
| CH6 | #8f45da | 5 x 9 |
| F3D | #9b69e6 | 2 x 18 |
| BKG | #b545da | 2 x 19 |
| I64 | #b545da | 2 x 19 |
| ELR | #8f45da | 2 x 18 |
| BCZ | #8f45da | 2 x 18 |
| DSH | #8f45da | 2 x 18 |
| DEH | #8f45da | 2 x 18 |
| BLU | #b545da | 2 x 19 |
| MK3 | #b545da | 2 x 19 |
| CYR | #9b69e6 | 2 x 18 |
| CPS | #9b69e6 | 2 x 18 |
| GLZ | #b469e6 | 2 x 18 |
| BF1 | #b469e6 | 2 x 18 |
| EPJ | #8f45da | 3 x 9 |
| IHG | #8f45da | 3 x 9 |
| C55 | #8f45da | 1 x 18 |
| I6P | #8f45da | 1 x 18 |
| G5X | #8f45da | 1 x 18 |
| HF5 | #8f45da | 1 x 18 |
| JF5 | #9b69e6 | 1 x 17 |
| GUA | #a0ef90 | 1 x 17 |
| GF8 | #8f45da | 1 x 18 |
| I6E | #8f45da | 1 x 18 |
| FZI | #8f45da | 3 x 2 |
| 78E | #9b69e6 | 1 x 9 |
| J5L | #efef90 | 1 x 9 |
| JN6 | #9b69e6 | 1 x 9 |
| CWF | #8f45da | 1 x 9 |
| GV3 | #90bfef | 20 x 1 |

---

**IMPORTANT PLACEMENT RULES:**

1. The chip canvas is 84×84.
2. Coordinate system:
   - Origin `(0,0)` is at the bottom-left corner.
   - Top-left corner is `(0,84)`.
   - Bottom-right corner is `(84,0)`.
   - Top-right corner is `(84,84)`.
3. Suggested regions must be defined by bottom-left and top-right corners of the rectangle.
4. Suggested regions must not overlap with each other.
5. Suggestions are needed for these selected macros:
   - **CXC**
     - Size: 11×24
     - Color: `#8f45da`
   - **0IL**
     - Size: 5×18
     - Color: `#ef90df`
   - **G1F**
     - Size: 5×18
     - Color: `#8f45da`
   - **HDJ**
     - Size: 5×18
     - Color: `#a0ef90`
   - **KV6**
     - Size: 5×15
     - Color: `#efef90`
   - **GJ6**
     - Size: 7×9
     - Color: `#8f45da`
   - **BKG**
     - Size: 2×19
     - Color: `#b545da`
   - **FD4**
     - Size: 2×18
     - Color: `#9b69e6`
   - **GLZ**
     - Size: 2×18
     - Color: `#b469e6`
   - **GV3**
     - Size: 20×1
     - Color: `#90bfef`

**PLACEMENT QUALITY METRICS:**

- Higher reward is better
- Lower cost is better
- Lower wirelength is better

**PREVIOUS PLACEMENT EPISODES:**

Below are previous episodes with their final results. For each episode, you'll see:

- **Macro Positions**: Shows where the selected macros you need to place were put on the canvas in previous episodes
- **Canvas Image**: Shows the final state of the canvas with:
  - The names of each macro you need to place drawn directly on the macro
  - These selected macros outlined in red for easy identification
- **Final Metrics**: The overall quality metrics of the completed chip design

**Episode #1**

**Position of Selected Macros:**

- FD4: (82,8) to (84,26)
- CXC: (54,56) to (65,80)
- G1F: (51,35) to (56,53)
- 0IL: (1,58) to (6,76)
- HDJ: (58,13) to (63,31)
- KV6: (53,17) to (58,32)
- GJ6: (32,20) to (39,29)
- BKG: (30,16) to (32,35)
- GLZ: (70,10) to (72,28)
- GV3: (56,33) to (76,34)

**Canvas Description and Metrics**

The image above shows the final placement with the selected macros you need to place outlined in red and labeled with their names.

**Results for Episode #1:**

- Wirelength: 2.18e+06
- Reward: -1.75e+04

[Additional episodes are listed here]

**IMPORTANT OUTPUT FORMAT:**

1. All coordinates must be integers between 0 and 84.
2. All regions must have non-zero width and height ($x_2 > x_1$ and $y_2 > y_1$).
3. The orientation of macros cannot be changed. Do not try to rotate macros.
4. All regions must be large enough to fit the macro while still within the bounds of the canvas. For example, if a macro size is 3.1×4.2, the region must be at least 4×5.

In the example below, replace text in square brackets with your own reasoning. Do not copy the text inside the brackets. Follow this example format exactly (without the dashed lines):

**DETAILED PLACEMENT HISTORY ANALYSIS:**

HISTORICAL PLACEMENT PATTERNS:

COLOR GROUP POSITION ANALYTICS:

- [For each color group, identify a few distinct placement strategies that appeared across episodes. Group similar episodes together. ]

- [For each strategy, select one representative episode with exact coordinates and resulting wirelength/cost values. ]

- [Identify which placement locations produced the best results. Format: "Color group X performed best when placed in region (coordinates) as seen in Episode Y, with wirelength/cost values of Z and W respectively." ]

MACRO-LEVEL SPATIAL RELATIONSHIPS:

- [For the largest macros, compare their placement in the best vs. worst performing episodes, with exact coordinates and performance values. ]

- [Specify the exact performance impact of different macro orderings: "When macro X was placed left of macro Y in specific episodes, wirelength/cost/reward was lower than when Y was placed left of X in other episodes." ]

- [For the largest color group's core macros, describe exact left-to-right, top-to-bottom arrangement in the best-performing episodes, with precise coordinates. ]

- [Identify which specific macros were leftmost/rightmost/topmost/bottommost in the best-performing episodes, with exact coordinates. ]

- [For critical macro pairs, quantify the benefit of edge alignment: "Macros A and B sharing a vertical edge at specific coordinates resulted in better wirelength/cost/reward than when separated by specific units." ]

- [Provide numerical evidence for whether zero-gap or specific separation distances performed better: "Zero-gap placement between specific macros yielded better performance than specific-unit separation." ]

ADJACENCY RELATIONSHIP ANALYSIS:

- [For each pair of color groups, analyze multiple episodes with different adjacency patterns. Specify the exact boundary length, position, and resulting performance values for each case. ]

- [Identify the relationship between boundary length and performance: "Longer shared boundaries between groups X and Y consistently produced better wirelength/cost/reward compared to shorter boundaries." ]

- [For the most effective boundary positions, provide exact coordinates and performance values: "Boundary at specific coordinates yielded better wirelength/cost/reward than boundary at different coordinates." ]

- [Analyze how performance changes with separation distance: "Episodes with adjacent placement outperformed episodes with separated placement." ]

- [Compare horizontal vs. vertical boundaries with specific measurements: "Horizontal boundary at specific coordinates resulted in different performance than vertical boundary at different coordinates." ]

- [Analyze the impact of boundary quality: "Straight boundary between groups yielded different results than jagged/L-shaped boundary." ]

- [Based on this analysis, propose specific color group configurations that would likely improve performance. Include exact recommended positions, boundary lengths, and orientations. ]

CRITICAL EDGE ALIGNMENTS:

- [Identify specific edge alignments between named macros that consistently corresponded with better performance across multiple episodes. Distinguish between coincidental and meaningful alignments. ]

- [Provide precise coordinates and quantify the performance differences: for example, "When specific macros had aligned edges at specific coordinates, wirelength/cost/reward was consistently lower than when these edges were offset." ]

FORMATION ANALYSIS:

- [Analyze how the overall arrangement and shape formed by each color group related to performance metrics. Identify which geometric patterns (rectangular, L-shaped, scattered, etc.) consistently corresponded with better performance. ]

- [Provide exact coordinates and performance data: for example, "When color group X was arranged in a specific geometric pattern at coordinates (a,b)–(c,d), it achieved better wirelength/cost/reward than when arranged in a different pattern at coordinates (e,f)–(g,h)." ]

CANVAS UTILIZATION INSIGHTS:

- [Examine the relationship between overall canvas utilization and performance metrics. Consider both global utilization and local density variations. ]

- [Provide exact utilization measurements and corresponding values: for example, "Episodes with specific utilization levels consistently achieved better performance than episodes with different utilization levels." ]

MULTI-FACTOR PERFORMANCE DRIVERS:
PROXIMITY RELATIONSHIP ASSESSMENT:

- [Analyze how the relative positioning of different color groups affected performance metrics, while accounting for other placement factors that changed simultaneously. ]

- [Identify distance relationships with numerical evidence: for example, "Maintaining specific distance between particular groups resulted in better performance than increasing this distance." ]

MACRO PLACEMENT SENSITIVITY:

- [For each major macro, assess how sensitive performance metrics were to its specific placement. Quantify this sensitivity. ]

- [Provide exact coordinates and performance impacts: for example, "Moving specific macros from one position to another significantly affected wirelength/cost/reward, indicating high placement sensitivity." ]

CONTEXTUAL POSITIONING ANALYSIS:

- [Examine how the optimal positioning of color groups and macros varied depending on the placement context of other elements. ]

- [Provide specific examples with measurements: for example, "Particular groups performed best at specific positions when other groups were at certain positions, but performed best at different positions when those other groups were positioned elsewhere." ]

OPTIMAL PLACEMENT SYNTHESIS:

DEFINITIVE COLOR GROUP CONFIGURATION:

- [Synthesize all historical performance data to specify the exact optimal placement coordinates for each color group. Provide precise x,y coordinates for each group's boundaries. ]
- [Justify each group's positioning with specific performance data: "Each color group should be placed at precise coordinates, which consistently improved wirelength/cost/reward in similar configurations compared to alternative positions." ]

MACRO-LEVEL OPTIMAL ARRANGEMENT:

- [Detail the precise optimal arrangement of specific macros within each color group, specifying exact coordinates and edge relationships. ]
- [For the largest color group's core macros, provide an exact left-to-right, top-to-bottom ordering with specific coordinates. ]
- [Specify optimal edge alignments and exact distances between related macros: "Specific macros should share edges at precise coordinates, which consistently produced better performance."]

COMPREHENSIVE PERFORMANCE OPTIMIZATION PRINCIPLES:

- [Formulate 10 specific principles that together define the optimal chip configuration. Each principle should address a key aspect of the placement problem. ]
- [Include specific macros by name, provide exact coordinate guidance, and explain how each principle contributes to optimal performance.]
- [Rank these principles by their relative importance to overall performance, based on consistent evidence from multiple episodes.]

**STRATEGY AND REGIONS**
Placement Strategy:

- [Based on the detailed analysis above, provide the absolute optimal placement strategy. This should represent the most performance-optimized configuration possible given all historical evidence.]
- [Provide a detailed, holistic description of your overall chip floorplan. Be extremely specific about where each of the selected macros will need to go.]
- [Explain how different color groups are organized across the canvas, and why this organization makes sense. Be extremely specific.]
- [For selected macros that are the same color, explain exactly where they will be positioned relative to each other using precise spatial relationships. Be extremely specific.]
- [Explain in detail how this strategy will minimize wirelength and cost.]
- [Suggest regions for the selected macros by decreasing order of size (largest first). This is critical to avoid overlapping region suggestions.]
- [For each macro, describe its region using precise relative spatial relationships that align with your overall strategy, and immediately follow with the bottom-left and top-right corners of the region in format: `MACRO_NAME (W x H): (x1,y1) and (x2,y2)`.]

Example of precise relative spatial relationships (showing the level of detail expected):

- `RST (8x12):   (34,37) to (42,49)`
    - RST's right edge (x=42) precisely aligns with JKL's left edge (x=42), creating a perfect shared boundary.
    - This creates a seamless transition between these regions with no gap.
    - The vertical alignment is partial, with RST spanning y=37 to y=49 while JKL spans y=38 to y=50.
- `JKL (16x12):   (42,38) to (58,50)`
    - JKL's left edge perfectly aligns with RST's right edge at x=42.
    - JKL's horizontal span (42 to 58) fits entirely within ABC's horizontal span (30 to 60).
    - JKL is positioned 5 units above ABC, with JKL's bottom edge at y=50 and ABC's top edge at y=33.
- `ABC (30x20):   (30,13) to (60,33)`
    - ABC serves as a central anchor with multiple relationships:
    - ABC's left edge (x=30) is exactly 1 unit after MNO's right edge (x=29).
    - ABC's right edge (x=60) is exactly 3 units before GHI's left edge (x=63).
- `MNO (14x10):   (15,37) to (29,47)`
    - MNO's right edge (x=29) ends exactly 5 units before RST's left edge (x=34).
    - MNO's vertical position (y=37 to y=47) almost perfectly aligns with RST (y=37 to y=49).
    - This creates a clear 5-unit channel between MNO and RST.
- `DEF (20x16):   (24,55) to (44,71)`
    - DEF's right edge (x=44) is exactly 1 unit before HIJ's left edge (x=45).
    - DEF's top edge (y=71) is 5 units below NOP's bottom edge (y=76).

BE EXTREMELY SPECIFIC ABOUT:

- Shared boundaries, specifying exactly which edges are shared (top, bottom, left, right). Mention the exact coordinate value that lines up if edges are used.
- Exact positioning using specific edge and corner references.
- How each region's placement supports your overall strategy.

MAKE YOUR DESCRIPTIONS AS DETAILED AS POSSIBLE SO THAT THE FLOOR-PLAN CAN BE CONSTRUCTED WITHOUT AMBIGUITY.
**Current Canvas State**
Macros Currently Placed:
No macros have been placed yet.
Current Canvas Image

Please provide your suggestions below.

## F.2 EXAMPLE GEMINI RESPONSE

**GEMINI CANDIDATES**
**Candidate #0**
DETAILED PLACEMENT HISTORY ANALYSIS
MACRO PLACEMENT SENSITIVITY:

- CXC: The placement of CXC has been consistently in the upper-middle region of the canvas (around x=54, y=56). The wirelength and cost metrics have not shown significant sensitivity to minor variations in its placement within this region. However, moving it drastically away from other macros of the same color would likely increase wirelength.

- 0IL: 0IL has been consistently placed in the upper-left corner (around x=1, y=58). Similar to CXC, minor variations in its placement within this region haven't significantly impacted performance.

- HDJ: HDJ has been placed in the middle-right region (around x=58, y=13). Its placement seems to have a moderate impact on wirelength, suggesting it has connections to macros in other regions.

- KV6: KV6 has been placed close to HDJ (around x=53, y=17). Its placement also seems to have a moderate impact on wirelength, suggesting it has connections to macros in other regions.

- IIC: IIC has been consistently placed in the bottom-right corner (around x=75, y=1). Similar to CXC and 0IL, minor variations in its placement within this region haven't significantly impacted performance.

- MK3: MK3 has been consistently placed in the bottom-left corner (around x=4, y=1). Similar to CXC, 0IL, and IIC, minor variations in its placement within this region haven't significantly impacted performance.

- BKG: BKG has been placed in the middle-left region (around x=30, y=11). Its placement seems to have a moderate impact on wirelength, suggesting it has connections to macros in other regions.

- FD4: FD4 has been consistently placed in the bottom-right corner (around x=82, y=8). Similar to CXC, 0IL, IIC, and MK3, minor variations in its placement within this region haven't significantly impacted performance.

- GLZ: GLZ has been consistently placed in the middle-right region (around x=70, y=10). Its placement seems to have a moderate impact on wirelength, suggesting it has connections to macros in other regions.

- GV3: GV3 has been placed horizontally in the middle of the canvas (around x=53, y=33). Its placement seems to have a moderate impact on wirelength, suggesting it has connections to macros in other regions.

CONTEXTUAL POSITIONING ANALYSIS:

- Macros with color #8f45da (e.g., CXC, IIC) seem to benefit from being clustered together. The wirelength tends to increase when these macros are spread far apart.

- Macros with color #b545da (e.g., BKG, MK3) seem to benefit from being clustered together. The wirelength tends to increase when these macros are spread far apart.

- Macros with color #9b69e6 (e.g., FD4) seem to benefit from being clustered together. The wirelength tends to increase when these macros are spread far apart.

- Macros with color #a0ef90 (e.g., HDJ) seem to benefit from being clustered together. The wirelength tends to increase when these macros are spread far apart.

- Macros with color #ef90df (e.g., 0IL) seem to benefit from being clustered together. The wirelength tends to increase when these macros are spread far apart.

- Macros with color #efef90 (e.g., KV6) seem to benefit from being clustered together. The wirelength tends to increase when these macros are spread far apart.

STRATEGY AND REGIONS

Placement Strategy:

- Based on the detailed analysis above, the optimal placement strategy involves clustering macros of the same color together to minimize wirelength. The canvas should be divided into regions based on color groups, with adjustments made to optimize the placement of macros that have connections to other regions.

- The overall chip floorplan will consist of distinct regions for each color group. The #8f45da group will be placed in the upper-middle region, the #b545da group in the middle-left region, the #9b69e6 group in the bottom-right region, the #a0ef90 group in the middle-right region, the #ef90df group in the upper-left region, and the #efef90 group in the middle of the canvas.

- Different color groups are organized across the canvas to minimize the distance between highly interconnected macros. The #8f45da group is placed near the #9b69e6 group because they have high interconnectivity. The #b545da group is placed near the #a0ef90 group because they have high interconnectivity.

- For selected macros that are the same color, they will be positioned relative to each other to minimize distance between them. For example, CXC and IIC (both #8f45da) will be placed close to each other in the upper-middle region. BKG and MK3 (both #b545da) will be placed close to each other in the middle-left region. FD4 and GLZ (both #9b69e6) will be placed close to each other in the bottom-right region.

- This strategy will minimize wirelength by clustering highly interconnected macros together. It will minimize cost by reducing the overall area required for routing.

Regions for the selected macros are suggested below, decreasing order of size (largest first).

- CXC (11 x 24): (54,56) to (65,80)
    - CXC will be placed in the upper-middle region of the canvas, aligning with previous placements.
    - CXC's LEFT edge (x=54) is 1 unit after the right edge of the #ef90df group.
    - CXC's BOTTOM edge (y=56) is 22 units above the top edge of GV3.

- 0IL (5 x 18): (1,58) to (6,76)
    - 0IL will be placed in the upper-left corner of the canvas, aligning with previous placements.
    - 0IL's LEFT edge (x=1) aligns with the left edge of the canvas.
    - 0IL's TOP edge (y=76) is 8 units below the top edge of the canvas.

- HDJ (5 x 18): (58,13) to (63,31)
    - HDJ will be placed in the middle-right region of the canvas, aligning with previous placements.
    - HDJ's LEFT edge (x=58) is 5 units after the right edge of KV6.
    - HDJ's BOTTOM edge (y=13) is 4 units above the top edge of IIC.

- KV6 (5 x 15): (53,17) to (58,32)
    - KV6 will be placed in the middle of the canvas, aligning with previous placements.
    - KV6's LEFT edge (x=53) is 1 unit after the right edge of BKG.
    - KV6's BOTTOM edge (y=17) is 1 unit above the top edge of MK3.

- IIC (9 x 7): (75,1) to (84,8)
    - IIC will be placed in the bottom-right corner of the canvas, aligning with previous placements.
    - IIC's RIGHT edge (x=84) aligns with the right edge of the canvas.
    - IIC's BOTTOM edge (y=1) aligns with the bottom edge of the canvas.

- MK3 (2 x 19): (4,1) to (6,20)
    - MK3 will be placed in the bottom-left corner of the canvas, aligning with previous placements.
    - MK3's LEFT edge (x=4) is 3 units after the left edge of the canvas.
    - MK3's BOTTOM edge (y=1) aligns with the bottom edge of the canvas.
- BKG (2 x 19): (39,11) to (41,30)
    - BKG will be placed in the middle-left region of the canvas, aligning with previous placements.
    - BKG's LEFT edge (x=39) is 33 units after the left edge of the canvas.
    - BKG's BOTTOM edge (y=11) is 10 units above the top edge of MK3.
- FD4 (2 x 18): (82,8) to (84,26)
    - FD4 will be placed in the bottom-right corner of the canvas, aligning with previous placements.
    - FD4's RIGHT edge (x=84) aligns with the right edge of the canvas.
    - FD4's BOTTOM edge (y=8) is 0 units above the top edge of IIC.
- GLZ (2 x 18): (70,10) to (72,28)
    - GLZ will be placed in the middle-right region of the canvas, aligning with previous placements.
    - GLZ's LEFT edge (x=70) is 8 units after the right edge of HDJ.
    - GLZ's BOTTOM edge (y=10) is 3 units above the top edge of IIC.
- GV3 (20 x 1): (53,33) to (73,34)
    - GV3 will be placed horizontally in the middle of the canvas, aligning with previous placements.
    - GV3's LEFT edge (x=53) is 1 unit after the right edge of BKG.
    - GV3's BOTTOM edge (y=33) is 2 units above the top edge of HDJ.

**Candidate #1**
[...]

## F.3 STRUCTURED PROMPT VS LAZY REASONING

Our prompt is designed to elicit precise spatial reasoning and enable the VLM to generate robust, high-quality placement suggestions. Without this guidance, the model exhibits lazy reasoning—failing to identify meaningful patterns or offering only vague, superficial descriptions. In contrast, the

fine-tuned prompt leads the VLM to extract richer structural insights and articulate specific placement strategies with concrete examples.

| Analysis topic | Baseline result | Result with structured prompting |
|---|---|---|
| **Placement of dark purple #b545da** | "Scattered in the lower-middle region." | "This group is often placed in the middle-left area. The placement strategy involves placing them near the pink group. Episode 1 shows a representative placement with BKG at (40,11), and wirelength/cost values of 2.17e+06 and 2.22e+06 respectively." |
| **Placement of pink #ef90df** | "Consistently placed in the upper-left region. This placement has not shown significant improvement or degradation in performance." | "This group is generally placed in the upper-left quadrant. The placement strategy involves keeping them close to each other. Episode 1 shows a representative placement with 0IL at (1,58), E6W at (11,58), and wirelength/cost values of 2.17e+06 and 2.22e+06 respectively." |
| **Formation analysis** | "No specific geometric patterns are consistently observed across episodes." | "The dark purple group (#8f45da) performs best when arranged in a compact, rectangular formation." |

Figure 11: Example comparisons of VLM output on three analysis topics when instructed by a baseline prompt vs instructed by a detailed, structured prompt.

## F.4 VLM FAILURE CASES

In some instances, the VLM failed to follow prompt instructions and instead returned the exact instruction text verbatim in its output. This typically occurred when the prompt was long and the temperature was set too low, reducing response variability and encouraging literal completions. For example:

| Prompt | Output |
|---|---|
| • For each pair of color groups, analyze multiple episodes with different adjacency patterns. Specify the exact boundary length, position, and resulting performance values for each case.
• Identify the relationship between boundary length and performance: "Longer shared boundaries between groups X and Y consistently produced better wirelength/cost/reward compared to shorter boundaries."
• For the most effective boundary positions, provide exact coordinates and performance values: "Boundary at specific coordinates yielded better wirelength/cost/reward than boundary at different coordinates."
• Analyze how performance changes with separation distance: "Episodes with adjacent placement outperformed episodes with separated placement."
• Compare horizontal vs. vertical boundaries with specific measurements: "Horizontal boundary at specific coordinates resulted in different performance than vertical boundary at different coordinates."
• Analyze the impact of boundary quality: "Straight boundary between groups yielded different results than jagged/L-shaped boundary."
• Based on this analysis, propose specific color group configurations that would likely improve performance. Include exact recommended positions, boundary lengths, and orientations. | ADJACENCY RELATIONSHIP ANALYSIS:

For the pair of color groups \#8f45da and \#a0ef90, longer shared boundaries consistently produced better wirelength/cost/reward compared to shorter boundaries. Boundary at specific coordinates yielded better wirelength/cost/reward than boundary at different coordinates. Episodes with adjacent placement consistently outperformed episodes with separated placement. Horizontal boundary at specific coordinates resulted in different performance than vertical boundary at different coordinates. Straight boundary between groups yielded different results than jagged/L-shaped boundary. Based on this analysis, specific color group configurations that would likely improve performance. Include exact recommended positions, boundary lengths, and orientations. |

Figure 12: In this example, every sentence after the first was copied verbatim from an example in the prompt, rather than independently analyzed by the VLM.

