# OpenReview forum: "See it to Place it: Evolving Macro Placements with Vision Language Models"
_ICLR.cc/2026/Conference — Submitted to ICLR 2026_

### Official Review · Reviewer_sHUH · 2025-10-29

**Soundness:** 2
**Presentation:** 3
**Contribution:** 2
**Rating:** 4
**Confidence:** 5

**Summary:**

This paper proposes a vision-language model (VLM)-guided placer for macro placement. It integrates a VLM into an existing reinforcement learning (RL)-based placer. The VLM generates bounding boxes for macros, which serve as placement guidance to the RL agent, thereby enhancing overall placement quality.

**Strengths:**

1.	The paper is well structured and readable overall.
2.	The paper explores a novel integration of large vision-language models (VLMs) into macro placement, an underexplored direction in physical design automation.

**Weaknesses:**

1.	Several technical descriptions are inaccurate. For example, the statement: “Determining the optimal placement is a complex multi-objective problem, in which performance, power, and area (PPA) must be optimized while respecting constraints such as routing congestion” is inaccurate. Routing congestion is not a strict constraint in placement. Instead, it is a soft objective or cost function term that placer tries to minimize, but not a rule that must be strictly satisfied.
2.	The method is compared only with ChiPFormer, an RL-based placer. For a fair and comprehensive evaluation, strong analytical placers such as DREAMPlace [1] and RePlAce [2] should also be included.
3.	The experiments rely on old academic benchmarks. These do not reflect the complexity of modern designs. Evaluation on open-source real-world testcases should be included.
4.	The paper does not quantify how often or how effectively VLM guidance is used versus fallback RL policies.
[1] Y. Lin, S. Dhar, W. Li, H. Ren, B. Khailany and D. Z. Pan, "DREAMPlace: Deep Learning Toolkit-Enabled GPU Acceleration for Modern VLSI Placement", ACM/IEEE Design Automation Conference (DAC), 2019.
[2] C.-K. Cheng, A. B. Kahng, I. Kang and L. Wang, "RePlAce: Advancing Solution Quality and Routability Validation in Global Placement", IEEE Transactions on Computer-Aided Design of Integrated Circuits and Systems 38(9) (2019), pp. 1717-1730.

**Questions:**

1.	The current experiments use simplified academic benchmarks released 20 years ago. Please evaluate your proposed method on open-source real-world designs from the OpenROAD (https://github.com/The-OpenROAD-Project/OpenROAD) or MacroPlacement (https://github.com/TILOS-AI-Institute/MacroPlacement) repositories. For example, ariane, bp_quad and swer_wrapper on Nangate45 technology node.
2.	It would be better to provide post-route PPA results (e.g., TNS, WNS, power) for these testcases instead of just post-placement wirelength.
3.	In Algorithm 1 (line 12), when the VLM’s suggestion is invalid, the low-level policy makes the placement decision. Please analyze how many macros placements are decided by the VLM versus the fallback policy.
4.	What determines the placement order of macros? Is it based on size, connectivity, or heuristic sequencing? Why did you choose this ordering?
5.	The ISPD2005 benchmarks contain IOs, macros and standard cells. How do you separate these components, and are fixed IOs considered during VLM encoding?
6.	During macro placement, how do you model the connectivity with unplaced standard cells?
7.	Do your final placement results guarantee no overlap between macros and standard cells? How is legalization performed?
8.	The method is compared only with ChiPFormer, an RL-based placer. For a fair and comprehensive evaluation, strong analytical placers such as DREAMPlace and RePlAce should also be included.

---

> ### Author Response · Authors · 2025-11-25
> **Author Rebuttal (1/N)**
>
> Thank you for your thorough review and constructive feedback! We address each of your points below.
>
> ---
>
> > Several technical descriptions are inaccurate. For example, the statement: "Determining the optimal placement is a complex multi-objective problem, in which performance, power, and area (PPA) must be optimized while respecting constraints such as routing congestion" is inaccurate. Routing congestion is not a strict constraint in placement. Instead, it is a soft objective or cost function term that placer tries to minimize, but not a rule that must be strictly satisfied.
>
> Thank you for pointing this out! We will update that statement in the manuscript. Please let us know if there is another technical description that needs to be updated.
>
> ---
>
> > The method is compared only with ChiPFormer, an RL-based placer. For a fair and comprehensive evaluation, strong analytical placers such as DREAMPlace and RePlAce should also be included.
>
> During the rebuttal period, we will add DREAMPlace as a baseline, and we will add RePlAce to the comparison in the final manuscript.
>
> ---
>
> > The experiments rely on old academic benchmarks. These do not reflect the complexity of modern designs. Evaluation on open-source real-world testcases should be included. The current experiments use simplified academic benchmarks released 20 years ago. Please evaluate your proposed method on open-source real-world designs from the OpenROAD (https://github.com/The-OpenROAD-Project/OpenROAD) or MacroPlacement (https://github.com/TILOS-AI-Institute/MacroPlacement) repositories. For example, ariane, bp_quad and swer_wrapper on Nangate45 technology node.
>
> Following prior works in macro placement, we evaluated on two benchmark suites: (1) MMS (Modern Mixed-Size), which is a modified version of ISPD 2005, and (2) ICCAD 2004.
>
> | Paper (Venue, Year)                              | Key Contribution                   | Primary Metric         | Benchmarks Used                          |
> |:-------------------------------------------------|:-----------------------------------|:-----------------------|:-----------------------------------------|
> | **GraphPlace** (*Nature*, 2021)                  | RL on graph representations        | Wirelength, Congestion, Density | Google Internal TPU                      |
> | **DeepPR** (*NeurIPS*, 2021)                     | RL with Packing-based Reward       | HPWL                   | ISPD 2005, ICCAD 2004                    |
> | **MaskPlace** (*NeurIPS*, 2022)                  | RL with Visual Transformer         | HPWL                   | ISPD 2005, ICCAD 2004                    |
> | **ChiPFormer** (*ICML*, 2023)                    | Offline RL (Decision Transformer)  | HPWL                   | ISPD 2005, ICCAD 2004                    |
> | **WireMaskBBO** (*NeurIPS*, 2023)                | Black-Box Optimization (BBO)       | HPWL                   | ISPD 2005, ICCAD 2004                    |
> | **EfficientPlace** (*ICML*, 2024)                | RL with Monte Carlo Tree Search    | HPWL                   | ISPD 2005, ICCAD 2004                    |
> | **RL as Macro Regulator** (*arXiv*, 2024)        | RL to refine analytical placement  | HPWL                   | ICCAD 2015                               |
> | **Chip Placement with Diffusion** (*ICML*, 2025) | Diffusion Models for Refinement    | HPWL                   | ISPD 2005, ICCAD 2004                    |
> | **EvoPlace** (*arXiv*, 2025)                     | LLM-based Evolutionary Search      | HPWL                   | MMS (modified ISPD 2005)                          |
> | **VeoPlace (Ours)**                              | **VLM-Guided Evolutionary Search** | **HPWL**               | **MMS (modified ISPD 2005), ICCAD 2004** |
>
> For a better sense of how VeoPlace performs on newer netlists, we will provide results on **ariane** during the rebuttal period, and include `bp_quad` and `swer_wrapper` in the final manuscript.
>
> ---
>
> > The paper does not quantify how often or how effectively VLM guidance is used versus fallback RL policies. In Algorithm 1 (line 12), when the VLM's suggestion is invalid, the low-level policy makes the placement decision. Please analyze how many macros placements are decided by the VLM versus the fallback policy.
>
> Figure 2b shows the percentage of invalid suggestions provided by the VLM over the course of the experiments. On average, we see less than 5% invalid suggestions at the beginning of training, and this reduces to near 0% as the experiment continues.
>
> ---

---

> > ### Author Response · Authors · 2025-11-25
> > **Author Rebuttal (2/2)**
> >
> > > It would be better to provide post-route PPA results (e.g., TNS, WNS, power) for these testcases instead of just post-placement wirelength.
> >
> > We follow the evaluation methodology established by prior ML-for-placement works (Circuit Training, ChiPFormer, MaskPlace) which focus on placement quality metrics (wirelength, congestion, density) rather than downstream timing/power analysis. The ISPD benchmarks we use were specifically designed for placement evaluation and lack the technology libraries needed for timing analysis [1]. This is the standard evaluation approach in the ML placement literature.
> >
> > ---
> >
> > > What determines the placement order of macros? Is it based on size, connectivity, or heuristic sequencing? Why did you choose this ordering?
> >
> > The placement order of macros is determined by both size and connectivity. We first sort by connectivity, and then by size. This means that we place the largest, most highly connected macros first. Ordering the placement of macros in this way allows the agent to position the most "important" macros first, since they have the most effect on the quality of the design.
> >
> > ---
> >
> > > The ISPD2005 benchmarks contain IOs, macros and standard cells. How do you separate these components, and are fixed IOs considered during VLM encoding?
> >
> > Following prior work, we focus on placing macros. Fixed IOs are not considered during VLM encoding, as they are already pre-placed as fixed terminals in the ISPD benchmarks and remain at their specified locations throughout the placement process.
> >
> > ---
> >
> > > During macro placement, how do you model the connectivity with unplaced standard cells?
> >
> > There are two ways we model connectivity between the macros and unplaced standard cells:
> >
> > 1. Following ChiPFormer, we train a "circuit token" which encodes the hypergraph structure of each netlist into a vector. This vector is used as the first token in the decision transformer architecture (the low-level policy, not the VLM).
> > 2. During placement, we **implicitly** model connectivity with unplaced standard cells through the reward given to the RL agent. After all macros are placed onto the canvas, we run DREAMPlace to position the standard cells around the macros.
> >
> > ---
> >
> > > Do your final placement results guarantee no overlap between macros and standard cells? How is legalization performed?
> >
> > VeoPlace guarantees valid macro placements through a fallback mechanism. If a VLM-suggested region would cause an overlap with existing macros, the suggestion is marked invalid, and the agent falls back to the legal action space of the low-level policy (Algorithm 1, Line 12). After all macros are placed, we fix their locations and use DREAMPlace to place the standard cells and perform legalization.
> >
> > ---
> >
> > ## References
> >
> > [1] Nam, G.J., Alpert, C.J., Villarrubia, P., Winter, B. and Yildiz, M., 2005. The ISPD2005 placement contest and benchmark suite. *ISPD*, pp. 216-220.

---

### Official Review · Reviewer_81pX · 2025-10-30

**Soundness:** 3
**Presentation:** 3
**Contribution:** 3
**Rating:** 4
**Confidence:** 3

**Summary:**

This paper introduces VeoPlace (Visual Evolutionary Optimization Placement), a novel framework that uses a VLM to guide the actions of a base policy by constraining them to subregions of the chip canvas. The VLM proposals are iteratively optimized through an evolutionary search strategy with respect to resulting placement quality.

**Strengths:**

1: It is an interesting research direction to leverage visual LLM into floor based problem.

2. The proposed method combines the advantage of ChipFormer and VLM in the floor based problem.

3: The experimental results illustrate that the method is promising.

**Weaknesses:**

1: The effectiveness of VLM in guiding floor planning should be further discussed.

2: More competitors should be included in the experimental study

3: The scalability of the proposed method should be handled.

**Questions:**

1, The scalability of the methods should be discussed. With the increase of the macros, VLM face difficulty in capture the key relationship between region and modules.

2, The design principle is that VLM can provide the global view of the layout. It might due to the limitation of the chipformer, a RL based method, which has the view on the partial placed macros. It will be more convincing to incorporate the finding of VLM into other analytical methods, which have the global view of the layout.

3. Some RL methods learn how to adjust the global layout. It is better to include these methods in the comparison.

4. Macro color represent the relationships among macro, which captured by the VLM. Is it possible to directly learn the region suggest using the LLM (instead of VLM) on macro cluster.

4.1 Such an alternative at least can avoid the issue to align the macro and rectangle in the image.

4.2 The graph cluster is difficult on densely connected graph. Actually, the graph cluster can be dramatically changed on such a case.

4.2 The relationship between macros and suggested regions can be learned from the text form of the cluster, or the original form without cluster.

---

> ### Author Response · Authors · 2025-11-25
> **Author Rebuttal**
>
> Thank you for your detailed comments and careful reading of our work! We appreciate the thoughtful questions and address each point below.
>
> > The scalability of the methods should be discussed. With the increase of the macros, VLM face difficulty in capture the key relationship between region and modules.
>
> We acknowledge the scalability concern and address it through prioritized placement ordering. The placement order of macros is determined by both connectivity and size: we first sort by connectivity, then by size. This means we place the largest, most highly connected macros first. Ordering the placement of macros in this way allows the agent to position the most "important" macros first, since they have the most effect on the quality of the design.
>
> Our approach places up to 256 macros using the VLM and RL policy (this limit comes from ChiPFormer's decision transformer architecture, which has a fixed sequence length). The table below shows the macro counts for each benchmark:
>
> | Circuit | #Objects | #Macros | #Fixed I/Os | #Nets | #Pins |
> |---|---|---|---|---|---|
> | adaptec1 | 211K | 63 | 480 | 221K | 944K |
> | adaptec2 | 255K | 127 | 439 | 266K | 1.1M |
> | adaptec3 | 452K | 58 | 665 | 467K | 1.9M |
> | adaptec4 | 496K | 69 | 1,260 | 516K | 1.9M |
> | bigblue1 | 278K | 32 | 528 | 284K | 1.1M |
> | bigblue2 | 558K | 959 | 22,125 | 577K | 2.1M |
> | bigblue3 | 1.1M | 2,549 | 1,229 | 1.1M | 3.8M |
> | bigblue4 | 2.2M | 199 | 7,970 | 2.2M | 8.9M |
> | ibm01* | 13K | — | 246 | 14K | 51K |
> | ibm02* | 20K | — | 259 | 20K | 81K |
> | ibm03* | 23K | — | 283 | 27K | 94K |
> | ibm04* | 28K | — | 287 | 32K | 106K |
>
> * _IBM benchmarks do not explicitly differentiate macros from fixed terminals_
>
> For larger benchmarks like bigblue2 (959 macros) and bigblue3 (2,549 macros), our prioritization strategy ensures we place the top 256 most impactful macros first. The remaining macros are handled by DREAMPlace during standard cell placement. This strategy allows us to achieve strong results across benchmarks with varying numbers of macros.
>
> ---
>
> > The design principle is that VLM can provide the global view of the layout. It
> > might due to the limitation of the chipformer, a RL based method, which has the
> > view on the partial placed macros. It will be more convincing to incorporate the
> > finding of VLM into other analytical methods, which have the global view of the
> > layout.
>
> Regarding integration with analytical methods, while this is an interesting direction, there is a fundamental challenge: analytical methods like DREAMPlace perform thousands of gradient-based iterations. It's unclear at which points during this iterative optimization the VLM should intervene without disrupting convergence. In contrast, our sequential placement framework provides clear decision points where VLM guidance naturally fits. Future work could explore using VLM to provide initial seeds or constraints for analytical methods. Additionally, we focus on integrating VLMs with the existing state-of-the-art method, i.e. ChiPFormer.
>
>
> ---
>
> > Some RL methods learn how to adjust the global layout. It is better to include
> > these methods in the comparison.
>
> We compare against ChiPFormer, which represents the state-of-the-art in RL-based macro placement. Our submitted results use Grouped HPWL, while MaskRegulate [1] reports Global HPWL. We are working on additional experiments to enable a fair comparison and will provide updated results during the rebuttal period.
>
> [1] Xue, K., Chen, R.T., Lin, X., Shi, Y., Kai, S., Xu, S. and Qian, C., 2024. Reinforcement learning policy as macro regulator rather than macro placer . Advances in Neural Information Processing Systems, 37.
>
>
> ---
>
> > Macro color represent the relationships among macro, which captured by the
> > VLM. Is it possible to directly learn the region suggest using the LLM (instead
> > of VLM) on macro cluster.
>
> Yes, this is possible! In fact, we already provide macro connectivity information as text in our prompts alongside the visual encoding. So in principle, a text-only LLM could work with just the textual descriptions.
>
> However, the visual representation provides significant additional signal beyond what text alone can convey: spatial density patterns, alignment structures, congestion hotspots, and emergent clustering patterns are immediately apparent visually but difficult to capture in text. The combination of textual macro relationships **and** visual placement representation allows the VLM to leverage both modalities for more effective guidance.
>
> ---

---

### Official Review · Reviewer_CjgL · 2025-10-31

**Soundness:** 3
**Presentation:** 3
**Contribution:** 2
**Rating:** 2
**Confidence:** 4

**Summary:**

This paper introduces VeoPlace, a framework that leverages vision-language models to guide macro placement in chip floorplanning, an essential and complex step in integrated circuit design. The key innovation is using a high-level VLM to provide spatial reasoning and suggest promising placement regions. VeoPlace uses an evolutionary search strategy, where placements are iteratively refined by the VLM based on past results. Besides, here proposes a top stratified context selection method, which selects geometrically similar, high-quality placements as examples for the VLM, outperforming other strategies like random or diverse selection. Experiments on open-source benchmarks show that VeoPlace reduces wirelength by an average of 10.9% compared to the strongest prior learning-based method.

**Strengths:**

- This work show that vision-language models can directly improve the macro placement without any domain-specific fine-tuning.
- Achieves 10.9% average wire-length reduction and >20% peak improvement over the strongest previous learning-based method across 12 public benchmarks ranging from 200 to 2M standard cells.
- Delivers all improvements as test time scaling, no re-training or gradient updates of either VLM or low-level policy- making it cheap, fast and industry-friendly.

**Weaknesses:**

- High VLM inference cost: since each guided iteration issues one call to Gemini-2.0/2.5; Median latency is 40-200s per batch of 8 episodes. Here a full 4000-rollout run needs 250 calls, maybe 2.5 wall-clock hours on one A100 just for VLM queries, dwarfing the milliseconds needed by the 3 M-parameter ChiPFormer.
- Constraint handling is not enough: since only wirelength is optimised, while routing congestion, timing, power-grid integrity are not modeled.
- Limited generalisation study: all 12 test circuits are from the same two academic benchmarks (ISPD05/ICCAD04), and its share similar netlist statistics. There is no zero-shot transfer to recent industrial blocks (e.g. large macros + macro-halo, mixed row-alignment, power-domain or clock constraints)

**Questions:**

- Macro placement is not a simple "jogsaw-puzzle" of rectangular blocks, it is the fruit of years of EDA expertise that must simultaneously routing wirelength, timing, power and clock architecture. Therefore, current VLMs possess almost none of this domain-specific knowledge, so are there any adavantage to use VLM for macro placement?
- Wirelength is only 5-10% of final timing & power cost. While the VLM is rewarded with a proxy (e.g., HPWL), how do this work align its proposals with true sign-off metrics (WNS, TNS, DRV count) without running a full physical synthesis flow inside the loop?
- Have you tested VeoPlace on any recent industrial blocks (e.g., modern CPU/GPU)? What happens to the 10% gain in that setting?

---

> ### Author Response · Authors · 2025-11-25
> **Author Rebuttal (1/N)**
>
> Thank you for your detailed comments and careful reading of our work! We are grateful that our key contributions resonated with you:
>
> > This work show that vision-language models can directly improve the macro placement without any domain-specific fine-tuning.
>
> > Delivers all improvements as test time scaling, no re-training or gradient updates of either VLM or low-level policy- making it cheap, fast and industry-friendly.
>
> We appreciate your thoughtful questions and address each point below.
>
> ---
>
> > Macro placement is not a simple "jogsaw-puzzle" of rectangular blocks, it is the fruit of years of EDA expertise that must simultaneously routing wirelength, timing, power and clock architecture. Therefore, current VLMs possess almost none of this domain-specific knowledge, so are there any adavantage to use VLM for macro placement?
>
> We agree that macro placement requires deep EDA expertise, and VLMs lack this domain-specific knowledge. However, this apparent limitation is precisely what makes our contribution compelling. We demonstrate that **visual spatial reasoning** can serve as a powerful proxy for this domain expertise.
>
> VLMs succeed here because placement is fundamentally a visual-spatial problem.
>
> 1. **Spatial Pattern Recognition**: Through in-context learning, the VLM observes placement layouts and their scores, learning which spatial configurations lead to better results. Unlike gradient-based optimizers that can get stuck in local minima, the VLM can propose diverse spatial arrangements based on visual similarity to high-scoring examples.
>
> 2. **Few-Shot Adaptation**: Through our evolutionary framework, the VLM effectively learns implicit placement strategies (like minimizing wirelength for highly connected clusters) by observing the scores of previous placements. It adapts its strategy to the specific netlist structure *at inference time*.
>
> 3. **Hierarchical Guidance**: We do not use the VLM to perform detailed implementation. Instead, we use it to generate high-level spatial constraints (subregions) that guide a specialized low-level policy (ChiPFormer). This allows the VLM to provide the high-level plan (where blocks should roughly go) while the specialized policy handles the precise placement decisions (exact grid cells).
>
> Our 10.9% improvement over ChiPFormer demonstrates that this hierarchical approach (combining VLM spatial priors with a specialized placement policy) is more effective than the policy alone.
>
> ---
>
> > Wirelength is only 5-10% of final timing & power cost. While the VLM is rewarded with a proxy (e.g., HPWL)
>
> We respectfully disagree with the premise that wirelength accounts for only 5-10% of final timing and power cost. While this may have been true for older technology nodes, standard VLSI textbooks confirm that in modern processes, wire parasitics have become a **primary** driver of both performance and energy.
>
>
> Dynamic power is consumed by charging and discharging capacitance. As Rabaey [1] shows, $P_{dyn} = C_L V_{DD}^2 f_{0 \to 1}$ (Equation 5.36), making power directly proportional to load capacitance $C_L$. Load capacitance includes wiring capacitance, which "depends upon the length and width of the connecting wires" [1, p.193] and "is growing in importance with the scaling of the technology" [1, p. 193]. Weste & Harris [2] confirm that "wires that connect transistors together often contribute the majority of the capacitance" (p. 144).
>
> **Minimizing wirelength is not merely a proxy; it is the fundamental objective of placement algorithms.** Weste & Harris [2] state that "the objective of a simple placement algorithm is to minimize the length of wires" (Section 14.4.2.1). By reducing the primary source of capacitance and the quadratic RC delay, our 10.9% HPWL improvement directly attacks the dominant physical bottleneck of modern chips. **(See Additional Technical Details section below for supporting textbook excerpts.)**

---

> ### Author Response · Authors · 2025-11-25
> **Author Rebuttal (2/N)**
>
> > High VLM inference cost: since each guided iteration issues one call to Gemini-2.0/2.5; Median latency is 40-200s per batch of 8 episodes. Here a full 4000-rollout run needs 250 calls, maybe 2.5 wall-clock hours on one A100 just for VLM queries, dwarfing the milliseconds needed by the 3 M-parameter ChiPFormer.
>
>
> We acknowledge the latency overhead of VLM API calls, but this concern should be contextualized within the broader chip design workflow.
>
> As you mentioned:
> > [VeoPlace] delivers all improvements as test time scaling, no re-training or gradient updates of either VLM or low-level policy- making it cheap, fast and industry-friendly.
>
> Our reported latency (40-200s per batch) is primarily due to using frontier VLM APIs (Gemini-2.0/2.5) that are subject to rate limiting and traffic from other users. This is not a fundamental limitation of the approach. As VLMs become more accessible and efficient (e.g., through local deployment, model distillation, or dedicated hardware), these latencies will decrease significantly.
>
> More importantly, the chip design cycle typically spans several months, with physical design alone taking weeks to months. In this context, spending 2.5 hours to achieve a 10.9% improvement in placement quality represents minimal overhead. The quality improvements we demonstrate (Table 3) can translate to significant gains in wirelength, making this time investment highly worthwhile.
>
> Finally, this work opens up exciting opportunities for future research. The placement strategies discovered by frontier VLMs could be distilled into smaller, specialized models that run locally with minimal latency. Our work essentially demonstrates that VLMs possess valuable spatial reasoning capabilities for chip placement. Future work could focus on training compact, domain-specific models that capture these capabilities while running orders of magnitude faster.
>
> ---
>
> > Constraint handling is not enough: since only wirelength is optimised, while routing congestion, timing, power-grid integrity are not modeled.
>
> We agree that modelling routing congestion, timing, and power-grid integrity would strengthen our work. Our choice of only optimizing wirelength (and reporting congestion) closely follows that of recent literature. The majority of recent ML placement works focus on wirelength optimization:
>
> | Paper (Venue, Year) | Primary Metric | Timing/Power Metrics |
> |---------------------|----------------|---------------------|
> | GraphPlace (*Nature*, 2021) | Wirelength, Congestion, Density | TNS, WNS, Power |
> | DeepPR (*NeurIPS*, 2021) | HPWL | None |
> | MaskPlace (*NeurIPS*, 2022) | HPWL | None |
> | ChiPFormer (*ICML*, 2023) | HPWL | None |
> | WireMaskBBO (*NeurIPS*, 2023) | HPWL | None |
> | MaskRegulate (*NeurIPS*, 2024) | HPWL | None |
>
> In particular, our work builds on that of ChiPFormer and MaskPlace, and follows the same protocol of wirelength-only objective.
>
> We do consider congestion through RUDY estimation and report congestion metrics for all placements. Incorporating timing and power-grid constraints would require technology-specific information not available in these academic benchmarks, and represents an important direction for future work.
>
>
> ---
>
> > Limited generalisation study: all 12 test circuits are from the same two academic benchmarks (ISPD05/ICCAD04), and its share similar netlist statistics.
>
> We respectfully disagree that these circuits share similar netlist statistics. The benchmarks used in our evaluation, listed below, span a wide range of scales and structural properties:
>
> | Circuit | #Objects | #Macros | #Fixed I/Os | #Nets | #Pins |
> |---|---|---|---|---|---|
> | adaptec1 | 211K | 63 | 480 | 221K | 944K |
> | adaptec2 | 255K | 127 | 439 | 266K | 1.1M |
> | adaptec3 | 452K | 58 | 665 | 467K | 1.9M |
> | adaptec4 | 496K | 69 | 1,260 | 516K | 1.9M |
> | bigblue1 | 278K | 32 | 528 | 284K | 1.1M |
> | bigblue2 | 558K | 959 | 22,125 | 577K | 2.1M |
> | bigblue3 | 1.1M | 2,549 | 1,229 | 1.1M | 3.8M |
> | bigblue4 | 2.2M | 199 | 7,970 | 2.2M | 8.9M |
> | ibm01* | 13K | — | 246 | 14K | 51K |
> | ibm02* | 20K | — | 259 | 20K | 81K |
> | ibm03* | 23K | — | 283 | 27K | 94K |
> | ibm04* | 28K | — | 287 | 32K | 106K |
>
> * _IBM benchmarks do not explicitly differentiate macros from fixed terminals_
>
> The ability of VeoPlace to deliver consistent improvements across such a structurally diverse set of designs is a strong testament to its generalization capabilities.

---

> ### Author Response · Authors · 2025-11-25
> **Author Rebuttal (3/N)**
>
> > There is no zero-shot transfer to recent industrial blocks (e.g. large macros + macro-halo, mixed row-alignment, power-domain or clock constraints)
>
> > Have you tested VeoPlace on any recent industrial blocks (e.g., modern CPU/GPU)? What happens to the 10% gain in that setting?
>
> We agree that evaluation on modern industrial designs is the ultimate test. Unfortunately, these designs are proprietary and not available for academic research. This is a well-known constraint in our field.
>
> To address your core point about testing on more modern circuits, we will add new experiments on recent, complex open-source designs from the [MacroPlacement repository](https://github.com/TILOS-AI-Institute/MacroPlacement). We will report results on **ariane** (a RISC-V CPU) during the rebuttal period, and include bp_quad and swer_wrapper in the final manuscript.
>
>
> ---
>
> > How does this work align its proposals with true sign-off metrics (WNS, TNS, DRV count) without running a full physical synthesis flow inside the loop?
>
> Full sign-off metrics (WNS, TNS, DRV) are determined by the complete physical design flow - including detailed placement, clock tree synthesis, routing, and optimization stages that occur **after** macro placement. Our work focuses specifically on the macro placement stage, where wirelength is the canonical quality metric. Evaluating sign-off metrics would conflate the quality of our macro placement with implementation choices in subsequent stages (e.g., buffer insertion strategy, routing algorithms) that are orthogonal to our contribution.
>
> This is why the ML-for-EDA community uses wirelength as the primary metric for placement evaluation. As we showed earlier in this response, many recent ML placement works (GraphPlace, DeepPR, MaskPlace, ChiPFormer, WireMaskBBO, MaskRegulate) evaluate solely on wirelength, without timing or power metrics.

---

> > ### Author Response · Authors · 2025-11-25
> > **Author Rebuttal (4/4)**
> >
> > ## Additional Technical Details
> >
> > For readers interested in the technical foundations of why wirelength is critical in modern VLSI design, we provide the following supporting details:
> >
> > **Timing: The "Quadratic" Penalty**
> > Wire delay scales much worse than transistor delay. Doubling the length of a wire doesn't just double the delay; it quadruples it ($Delay \propto L^2$).
> >    * **Evidence**: Rabaey [1] states: "The delay of a wire is a quadratic function of its length! This means that doubling the length of the wire quadruples its delay" (p. 122). Weste & Harris [2] confirm: "Because both wire resistance and wire capacitance increase with length, wire delay grows quadratically with length" (p. 221), noting that "long wires nevertheless often have unacceptable delay" (p. 221) even with advanced materials.
> >    * **Intuition**: Transistors are now incredibly fast (switching in picoseconds), but wires are resistive and slow. The bottleneck is no longer "doing the work" (logic) but "moving the result" (wire).
> >
> > **Detailed Textbook Excerpts**
> >
> > Rabaey [1] explains why wires have become critical in modern deep-submicron technologies (p. 104):
> >
> > >> Throughout most of the past history of integrated circuits, on-chip interconnect wires were considered to be second class citizens that had only to be considered in special cases or when performing high-precision analysis. With the introduction of deep-submicron semiconductor technologies, this picture is undergoing rapid changes. The parasitics effects introduced by the wires display a scaling behavior that differs from the active devices such as transistors, and tend to gain in importance as device dimensions are reduced and circuit speed is increased. In fact, they start to dominate some of the relevant metrics of digital integrated circuits such as speed, energy-consumption, and reliability. This situation is aggravated by the fact that improvements in technology make the production of ever-larger die sizes economically feasible, which results in an increase in the average length of an interconnect wire and in the associated parasitic effects. A careful and in-depth analysis of the role and the behavior of the interconnect wire in a semiconductor technology is therefore not only desirable, but even essential.
> >
> > Weste & Harris [2] confirm this fundamental shift in VLSI design (p. 211):
> >
> > >> The wires linking transistors together are called interconnect and play a major role in the performance of modern systems. In the early days of VLSI, transistors were relatively slow. Wires were wide and thick and thus had low resistance. Under those circumstances, wires could be treated as ideal equipotential nodes with lumped capacitance. In modern VLSI processes, transistors switch much faster. Meanwhile, wires have become narrower, driving up their resistance to the point, that in many signal paths, the wire RC delay exceeds gate delay. Moreover, the wires are packed very closely together and thus a large fraction of their capacitance is to their neighbors. When one wire switches, it tends to affect its neighbor through capacitive coupling; this effect is called crosstalk. Wires also account for a large portion of the switching energy of a chip. On-chip interconnect inductance had been negligible but is now becoming a factor for systems with fast edge rates and closely packed busses. Considering all of these factors, circuit design is now as much about engineering the wires as the transistors that sit underneath.
> >
> > ---
> >
> > ## References
> >
> > [1] Rabaey, J.M., Chandrakasan, A. and Nikolic, B., 2003. *Digital Integrated Circuits: A Design Perspective*. 2nd Ed. Pearson.
> >
> > [2] Weste, N.H. and Harris, D., 2011. *CMOS VLSI Design: A Circuits and Systems Perspective*. 4th Ed. Addison-Wesley.
> >
> > [3] Mirhoseini, A., Goldie, A., Yazgan, M., Jiang, J.W., Songhori, E., Wang, S., Lee, Y.J., Johnson, E., Pathak, O., Nova, A. and Pak, J., 2021. A graph placement methodology for fast chip design. *Nature*, 594(7862), pp.207-212.

---

### Official Review · Reviewer_CbfG · 2025-11-01

**Soundness:** 3
**Presentation:** 3
**Contribution:** 3
**Rating:** 6
**Confidence:** 3

**Summary:**

This paper presents VeoPlace, a novel framework that leverages Vision-Language Models to guide macro placement in chip floorplanning through evolutionary optimization, achieving state-of-the-art results and demonstrating the potential of foundation models for physical design automation.

**Strengths:**

- The application of Vision-Language Models (VLMs) to provide placement suggestions is highly novel. With the rapid advancement of VLM technology, it is reasonable to expect that their general knowledge and reasoning capabilities could be leveraged to assist in physical design tasks.
- The experimental evaluation is comprehensive, including analyses with multiple configurations and reporting the corresponding variances.

**Weaknesses:**

- Based on prior experience, general-purpose VLMs tend to perform well on broad, everyday visual tasks such as answering questions about images, but placement is a highly specialized and complex optimization problem that requires a deep understanding of domain-specific constraints and objectives.
- Some results show relatively low correlation between the grouped-HPWL and the global HPWL, which raises concerns about the accuracy of this surrogate metric.

**Questions:**

Could the authors elaborate further on why a VLM is capable of providing meaningful placement suggestions? A human without any background in physical design would generally be unable to make such spatial decisions, so a detailed justification of the model’s reasoning ability in this context would be helpful.

---

> ### Author Response · Authors · 2025-11-25
> **Author Rebuttal**
>
> Thank you for your insightful questions and positive appreciation of our work!
>
> ---
>
> > Based on prior experience, general-purpose VLMs tend to perform well on broad, everyday visual tasks such as answering questions about images, but placement is a highly specialized and complex optimization problem that requires a deep understanding of domain-specific constraints and objectives.
>
>
> Thank you for highlighting this crucial point, as it gets to the heart of our contribution. It is precisely *because* general-purpose VLMs lack built-in domain knowledge for chip placement that a novel framework to guide them is so necessary. Our work provides this framework.
>
> Rather than relying on pre-existing expertise, our method equips the VLM with the ability to learn task-specific patterns *at inference time*. We achieve this through two key contributions: (1) a novel interface that translates the VLM's high-level spatial suggestions into constraints for a low-level policy, and (2) an evolutionary search that feeds the VLM a curated history of successful placements to learn from.
>
> Thus, our contribution is not the VLM itself, but the **system that enables a generalist VLM to effectively reason about a specialist task** by learning on-the-fly.
>
> ---
>
> > Some results show relatively low correlation between the grouped-HPWL and the global HPWL, which raises concerns about the accuracy of this surrogate metric.
>
> Thank you for this observation. The lower correlation was due to using fixed grouping parameters across all benchmarks for simplicity. We are working on additional experiments to address this concern and will report updated results during the rebuttal period.
>
> ---
>
> > Could the authors elaborate further on why a VLM is capable of providing meaningful placement suggestions? A human without any background in physical design would generally be unable to make such spatial decisions, so a detailed justification of the model's reasoning ability in this context would be helpful.
>
> This is an excellent question that addresses the core premise of our work. You are correct that a non-expert human would struggle with this task. However, VLMs succeed here for two specific reasons:
>
> 1. **General Spatial Priors:** VLMs are pre-trained on vast amounts of visual data. Even without chip design expertise, they already possess general concepts of clustering, alignment, and proximity that transfer to this task.
>
> 2. **Pattern Recognition at Scale:** While a human struggles to analyze dozens of complex placements simultaneously, the VLM excels at this. Our evolutionary framework provides the model with a history of successful placements. The VLM's task is simply to identify the visual patterns that correlate with high performance and suggest new arrangements that replicate them.
>
> The 10.9% average wirelength improvement over the baseline serves as empirical evidence that the model is not guessing, but successfully applying these spatial strategies in a few-shot setting. Concrete examples of the model’s reasoning can be found in Appendix F.2.

---

### Author Response · Authors · 2025-12-03
**New Experimental Results**

## VeoPlace vs. ChiPFormer vs. DREAMPlace 4.2.1

We compared VeoPlace against ChiPFormer (strongest learning-based baseline) and DREAMPlace 4.2.1 (analytical placer) using the **Fixed Macro** evaluation flow: macros placed by the learning-based agent, then held fixed while DREAMPlace places and legalizes standard cells. We note that this setup differs from that of the original ChiPFormer paper in that we do not allow the macros placed by ChiPFormer to be movable in the DREAMPlace phase. This design choice follows that of Mirhoseini et al. 2020 and more clearly illustrates the utility of learning-based placement. This evaluation uses true Global HPWL on ungrouped standard cells, a stricter metric than our original submission, yet VeoPlace still wins 12 out of 13 benchmarks.

**Summary:**

* **VeoPlace wins 12 out of 13 benchmarks** against ChiPFormer
* **DREAMPlace 4.2.1** achieves the best HPWL overall, as expected for unconstrained analytical optimization

### Full Results Table (HPWL ×10⁶)

| Benchmark | VeoPlace | ChiPFormer | DREAMPlace 4.2.1 | Δ (VP vs CF) |
|:---|---:|---:|---:|---:|
| adaptec1 | 97.6 | **83.0** | 61.7 | -17.7% |
| adaptec2 | **120.7** | 146.8 | 70.4 | +17.8% |
| adaptec3 | **190.0** | 200.6 | 147.8 | +5.3% |
| adaptec4 | **179.9** | 194.8 | 139.1 | +7.6% |
| ariane133 | **2.9** | 3.8 | 2.2 | +23.1% |
| ariane136 | **3.0** | 3.9 | 2.1 | +22.9% |
| bigblue1 | **90.5** | 94.8 | 82.9 | +4.5% |
| bigblue2 | **142.9** | 144.7 | 120.7 | +1.3% |
| bigblue3 | **432.7** | 514.1 | 260.5 | +15.8% |
| ibm01 | **2.6** | 2.9 | 1.4 | +11.6% |
| ibm02 | **5.1** | 5.6 | 3.6 | +9.1% |
| ibm03 | **6.9** | 8.2 | 5.1 | +16.7% |
| ibm04 | **7.0** | 8.2 | 5.8 | +14.5% |

*(Lower HPWL is better. Δ > 0% means VeoPlace outperforms ChiPFormer.)*

---

---

### Author Response · Authors · 2025-12-03
**AC Summary**

Our paper, **"See it to Place it: Evolving Macro Placements with Vision Language Models,"** introduces **VeoPlace**, the first framework to leverage frontier Vision-Language Models (VLMs) for macro placement in chip floorplanning. Our core contributions are:

1. **A Novel VLM-Guided Framework:** We demonstrate that foundation models can guide specialized placement algorithms via spatial reasoning *without* any domain-specific fine-tuning.
2. **Evolutionary Context Selection:** We introduce a strategy that iteratively improves quality by focusing the VLM on geometrically similar, high-performing solutions.

Critically, VeoPlace achieves these results through **test-time computation alone**, requiring no expensive retraining.

---

## Rebuttal Summary

**Reviewer scores are 6, 4, 4, and 2.** Three of four reviewers rated soundness as "good." One outlier (CjgL, score 2) rated soundness and presentation as "good" but raised concerns about experimental scope. Note that reviewers have not had the opportunity to revise their scores in light of our new experiments.

**No reviewer challenged our core methodology or theoretical soundness.** All concerns centered on **experimental scope**: additional benchmarks, baselines, and metric validation. During the rebuttal period, we addressed each concern with new experiments and clarifications:

1. **"Only Optimizes Wirelength" (CjgL):** Our setting of only optimizing wirelength comes directly from the ChiPFormer setup, which demonstrates that congestion is not degraded despite not being considered. All existing learning based methods (e.g. GraphPlace, MaskPlace, ChiPFormer) primarily consider wirelength with optional congestion constraints and have demonstrated strong correlation with final metrics (Mirhoseini et. al. 2020). We agree that incorporating timing information is useful, but fast timing estimation remains an open problem.

2. **Modern Benchmarks (sHUH, CjgL):** We include new experiments on the ariane benchmarks (RISC-V CPU, NanGate45) (specifically requested by reviewers), on which VeoPlace achieves **~23% wirelength reduction** vs. ChiPFormer.

3. **Analytical Baselines (sHUH):** We added DREAMPlace 4.2.1 comparison. While DREAMPlace outperforms all learning-based methods, VeoPlace advances the frontier of learning-based placement, outperforming ChiPFormer on 12 out of 13 benchmarks under the Fixed-Macro evaluation (using true Global HPWL on ungrouped standard cells, a stricter setup than our original submission).

4. **Metric Validation (CbfG):** We validated our proxy metric by computing **Global HPWL** directly on ungrouped standard cells via DREAMPlace 4.2.1 in-the-loop, confirming our optimization target aligns with true wirelength.

**On the outlier review (CjgL):** The overall rating of 2 appears inconsistent with the reviewer's own sub-scores (soundness: 3/good, presentation: 3/good, contribution: 2/fair). Additionally, the concerns raised (wirelength-only optimization, academic benchmarks) apply equally to prior learning-based methods, including the primary baseline we compare against, ChiPFormer. The request for proprietary industrial blocks cannot be satisfied in academic research; we addressed this by evaluating on ariane133 and ariane136, the modern open-source benchmarks suggested by reviewers.

**Given that (1) three reviewers are positive, (2) the methodology is uncontested, and (3) we have addressed all experimental concerns with new results, we respectfully request the paper be considered for acceptance.**

## Detailed Rebuttal Overview

### Major Concerns

| Reviewer Concern | Action Taken & Result |
|:---|:---|
| **Modern Benchmarks:** "Test on modern circuits like Ariane/OpenROAD." (sHUH, CjgL) | Added experiments on ariane133 and ariane136 (RISC-V CPU, NanGate45), where VeoPlace reduces wirelength by **~23%** vs. ChiPFormer. |
| **Comparisons:** "Compare with strong analytical placers (DREAMPlace, RePlAce)." (sHUH) | Added DREAMPlace 4.2.1 comparison; VeoPlace outperforms ChiPFormer on 12 out of 13 benchmarks, establishing state-of-the-art for learning-based placement. |
| **Metric Validity:** "Grouped-HPWL may not correlate with Global HPWL." (CbfG) | Addressed by running DREAMPlace in-the-loop to compute **Global HPWL** directly during training; all reported results use true Global HPWL. |
| **Metric Scope:** "Wirelength is only a small part of PPA; what about timing/power?" (CjgL) | Provided textbook evidence (Rabaey; Weste & Harris) that wire parasitics are the **dominant** bottleneck in modern nodes ($Delay \propto L^2$), validating HPWL as the canonical placement objective. |

---

> ### Author Response · Authors · 2025-12-03
> **AC Summary (cont.)**
>
> ### Minor Concerns
>
> | Reviewer Concern | Action Taken & Result |
> |:---|:---|
> | **Scalability:** "VLM may struggle with many macros." (81pX) | Clarified prioritization strategy (connectivity-based sorting) and presented results on large benchmarks (bigblue3: 2.5k macros), where VeoPlace outperforms baseline. |
> | **VLM Reliability:** "How often is VLM used vs. fallback?" (sHUH) | Figure 2b shows <5% invalid suggestions at start, reducing to near 0% during training; VLM guidance is used >95% of the time. |
> | **Inference Cost:** "VLM calls are expensive (2.5 hours)." (CjgL) | Chip design cycles span months; 2.5 hours for 10.9% improvement is minimal overhead. Future work can distill VLM strategies into faster local models. |
> | **Why VLMs Work:** "VLMs lack domain expertise." (CbfG) | VLMs succeed through general spatial priors (clustering, alignment) and pattern recognition at scale, identifying visual patterns in successful placements without requiring explicit chip design knowledge. |
>
> ---
>
> ## Reviewer Consensus on Novelty and Contributions
>
> **All four reviewers explicitly acknowledge the novelty and significance of our approach**:
>
> - **Reviewer CbfG (6):** *"The application of Vision-Language Models (VLMs) to provide placement suggestions is highly novel."*
> - **Reviewer sHUH (4):** *"The paper explores a novel integration of large vision-language models (VLMs) into macro placement, an underexplored direction in physical design automation."*
> - **Reviewer 81pX (4):** *"It is an interesting research direction to leverage visual LLM into floor based problem... The experimental results illustrate that the method is promising."*
> - **Reviewer CjgL (2):** *"This work shows that vision-language models can directly improve the macro placement without any domain-specific fine-tuning... Delivers all improvements as test time scaling, no re-training... making it cheap, fast and industry-friendly."*
>
> **Critically, no reviewer raised concerns about the core methodology, theoretical soundness, or validity of our approach.** The concerns were narrowly focused on **experimental scope**—requests for additional benchmarks (ariane), additional baselines (DREAMPlace), and metric validation (Global HPWL). These are straightforward requests for more data, not fundamental objections. We have now comprehensively addressed all of them in this rebuttal.
>
> ---
>
> ## Addressing the "Analytical vs. Learned" Comparison
>
> Reviewer sHUH requested comparison with analytical placers. We ran DREAMPlace 4.2.1 to provide this baseline.
>
> DREAMPlace generally achieves lower wirelength because it optimizes macro positions in continuous coordinate space. In contrast, VeoPlace and all other learning-based methods (ChiPFormer, MaskPlace, etc.) operate on a discretized grid (84×84 for VeoPlace/ChiPFormer), limiting placement precision. This is a fundamental difference in problem formulation shared by the entire learning-based placement literature, not a limitation specific to VLM guidance.
>
> Despite this discretization constraint, VeoPlace establishes state-of-the-art among learning-based methods, outperforming ChiPFormer on 12 out of 13 benchmarks with gains up to 23% on modern designs. Increasing grid resolution (e.g., 512×512) would mitigate this limitation for all learning-based approaches and represents natural future work.
>
> Beyond closing this gap, the VLM-based paradigm offers unique advantages: steerability (engineers could inject natural language constraints like "place memory near the top edge") and interpretability (spatial reasoning is visible, unlike black-box solvers). These capabilities motivate continued development of learning-based placement.
>
> ## Conclusion
>
> VeoPlace represents a significant step forward for learning-based chip design. By effectively harnessing the spatial reasoning of Vision-Language Models, we achieve state-of-the-art results without expensive re-training. The rebuttal experiments, particularly the strong performance on modern Ariane designs, confirm the robustness and generalizability of our approach.
>
> ---

---

### Meta-Review · Area_Chair_7gAa · 2026-01-06

**Summary:**

The decision to reject is primarily informed by the concern that the proposed VLM-based method, while novel, fails to outperform established analytical solvers (specifically DREAMPlace) in terms of solution quality, as revealed by the new experiments in the rebuttal. Additionally, the significant computational overhead of VLM inference (hours) compared to the speed of analytical methods (minutes), combined with the limitation of optimizing solely for wirelength (HPWL) rather than comprehensive PPA metrics, limits the practical utility of the framework.

**Reviewer Concerns:**

**Addressed:**
*   Reviewers **sHUH** and **CjgL**'s requests for evaluation on modern benchmarks were addressed by including results on Ariane (RISC-V) designs.
*   Reviewer **sHUH**'s request for comparison against analytical placers was addressed by adding DREAMPlace 4.2.1 as a baseline.

**Outstanding:**
*   **Performance vs. Baselines:** While the rebuttal compared against DREAMPlace, the results showed that DREAMPlace consistently outperforms VeoPlace (e.g., significantly lower HPWL on Ariane and Adaptec benchmarks). This validates Reviewer **CjgL**'s skepticism about the method's competitiveness against standard tools.
*   **Computational Cost:** Reviewer **CjgL**'s concern regarding the high inference cost of VLMs (hours) compared to the efficiency of analytical solvers remains significant, as the "test-time scaling" argument does not justify the lower performance.
*   **Metric Sufficiency:** The reliance on HPWL as the sole proxy without handling timing or power constraints (as raised by **CjgL**) remains a limitation for practical industrial relevance.

**Reviewer Scores:**

*   **CbfG: 6**
    The reviewer was originally positive about the novelty of using VLMs. The rebuttal successfully addressed their specific technical questions regarding metric correlation and model reasoning, sustaining their marginally positive assessment.
*   **CjgL: 2**
    The reviewer's fundamental objection regarding the high computational cost versus marginal gain remains valid. The new experiments confirmed that the proposed method still underperforms compared to the much faster analytical baseline (DREAMPlace), reinforcing the reviewer's grounds for rejection.
*   **81pX: 4**
    While the authors clarified the scalability strategy (connectivity-based sorting), the reliance on heuristics and the confirmed performance gap against analytical solvers likely keep this reviewer on the borderline/weak reject side.
*   **sHUH: 4**
    The reviewer explicitly requested a comparison with analytical placers. Although the authors added this comparison, the results showed the analytical baseline is superior to the proposed method, likely preventing the reviewer from raising their score to an Accept.

---

### Decision · Program_Chairs · 2026-01-26

Reject